# A Review on Binder Jet Additive Manufacturing of 316L Stainless Steel

**Saereh Mirzababaei** [1,2] and **Somayeh Pasebani** [1,2,*]

[1]   School of Mechanical, Industrial and Manufacturing Engineering, Oregon State University,
      Corvallis, OR 97330, USA

[2]   Advanced Technology and Manufacturing Institute (ATAMI), Corvallis, OR 97330, USA

[*]   Correspondence: somayeh.pasebani@oregonstate.edu; Tel.: +1-541-737-3685

**Abstract:** Binder jet additive manufacturing enables the production of complex components for numerous applications. Binder jetting is the only powder bed additive manufacturing process that is not fusion-based, thus manufactured parts have no residual stresses as opposed to laser-based additive manufacturing processes. Binder jet technology can be adopted for the production of various small and large metallic parts for specific applications, including in the biomedical and energy sectors, at a lower cost and shorter lead time. One of the most well-known types of stainless steels for various industries is 316L, which has been extensively manufactured using binder jet technology. Binder jet manufactured 316L parts have obtained near full density and, in some cases, similar mechanical properties compared to conventionally manufactured parts. This article introduces methods, principles, and applications of binder jetting of SS 316L. Details of binder jetting processes, including powder characteristics (shape and size), binder properties (binder chemistry and droplet formation mechanism), printing process parameters (such as layer thickness, binder saturation, drying time), and post-processing sintering mechanism and densification processes, are carefully reviewed. Furthermore, critical factors in the selection of feedstock, printing parameters, sintering temperature, time, atmosphere, and heating rate of 316L binder jet manufactured parts are highlighted and summarized. Finally, the above-mentioned processing parameters are correlated with final density and mechanical properties of 316L components to establish a guideline on feedstock selection and process parameters optimization to achieve desired density, structure and properties for various applications.

**Keywords:** additive manufacturing; binder jetting; three-dimensional printing; stainless steel (SS) 316L; powder bed; sintering; diffusion; porosity; densification; mechanical properties

---

## 1. Introduction

Additive manufacturing (AM) is getting more attention because of manufacturing complex geometric parts directly from computer-aided design (CAD) files that cannot be produced by traditional manufacturing processes at a significantly reduced cost, energy, and material consumption, chemical waste, process steps, and human resources [1–8]. AM processes selectively add materials layer-by-layer, to build apart from three-dimensional (3D) models [9–12].

AM technology potentially affects machinery, assembly processes, and supply chains in traditional production models [11,13]. Seven categories for AM technology are defined by the American Society for Testing and Materials (ASTM): vat photopolymerization, material jetting, binder jetting, powder bed fusion, material extrusion, sheet lamination, and direct energy deposition [5,8,9,14–16]. AM technology offers the creation of parts from various range of materials, including metals, ceramics, polymers, sand, and glass [5,8–12,17–19].

Material extrusion uses polymer filaments as feedstock material [5,20,21], sheet lamination utilizes sheets, and direct energy deposition uses wires or powder. Binder jetting (BJ), powder bed fusion (PBF), and electron beam melting (EBM) are powder bed-based methods [5,22,23] in supplying for rapid prototyping demands and building parts with complex internal features that could not be manufactured by conventional powder metallurgy (press and sinter) processes [24,25]. In contrast to other powder bed processes that apply laser or electron beam to fuse powder particles, no heat source or fusion is involved in the binder jet processes. Instead, liquid binder joins metal powder particles together. This is a significant advantage of binder jetting over other powder bed-based processes because an unlimited range of materials can be manufactured at room temperature and room atmosphere for many industrial applications.

Binder jetting known as 3D printing (3DP) was developed at MIT in the early 1990s [14,16,26]. Binder jet method selectively prints a binder on the powder bed to join the powders layer-by-layer and, form a green part [14,15,26–29]. Binder jet processes include seven steps: Printing, curing, de-powdering, sintering, infiltration, annealing, and finishing [14,15,30]. First, BJ software slices the given 3D CAD file to 2D files corresponding to defined layer thickness. The 2D files are used as the printer's input. Binder jet printing process shown in Figure 1, begins with spreading a thin layer of powder in the print bed. Then, the print head passes over the platform and deposits binder onto the powder only on relevant spots based on 2D file information, thus building the first layer. When the first layer is printed, the printing system lowers the build box and lifts the supply bed by one-layer thickness. A new layer of powder is spread, and the binder is printed on it. The printing process is repeated layer-by-layer until the part is complete [15,27].

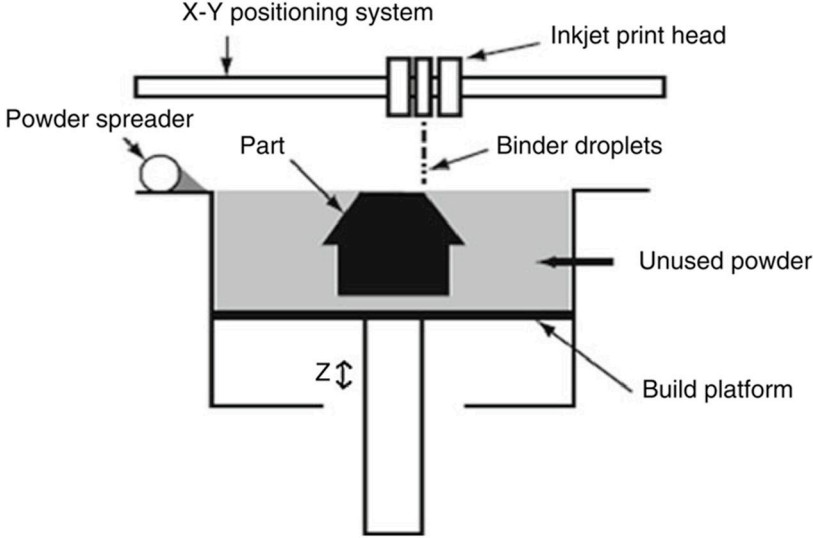

**Figure 1.** Schematic of binder jetting printing process [26].

Unlike PBF processes which apply laser or electron beam to melt metal powders, no power heat source involved during the BJ building process, thus BJ can work with a wide range of metals and is regarded as one of the most cost-effective AM techniques [5,12,14,26,31,32]. Furthermore, the surrounding powder sufficiently supports any overhanging geometry in the powder bed and eliminates the need for support structure design [5,27]. BJ printing process can be accelerated by increasing the number of print head nozzles [5,12]. Since the printing process performs at room temperature, no residual stress or distortion is introduced to the part due to thermal gradients [4,5,12,27]. Unbonded powders are fully recyclable so that can be used [26]. However, the lack of local melting increases the possibility for the presence of porosity and hinders attaining a high relative density. Before sintering, the relative density of metallic parts created by BJ is typically around 50–60% of the theoretical density which is noticeably lower than the green compacts pressed in powder metallurgy

(P/M) process [4,5]. Pores act as high-stress concentration areas promoting crack propagation. Thus, reducing porosity in the printed part can significantly improve the mechanical properties [12]. After printing, binder curing is required to improve the strength of the green part. To remove the part from powder bed and excessive surrounding powder, a soft brush is generally used. Sintering is the most crucial post-processing step that improves part density and strength while burns off the binder and densifies the part. During sintering, porosity reduction and dimensional shrinkage occur that will significantly affect the dimensional accuracy and part quality [14,15,26].

Generally, the quality of the part highly depends on the process parameters. Although BJ has many advantages, to be widely adopted in all industries, including automotive, aerospace, medical, and biomedical, the optimal BJ process needs to be identified. Furthermore, density, microstructure, and mechanical properties require further improvement to become competitive with conventionally manufactured parts in terms of quality, performance, durability, cost, and production time [12,15].

## 2. Current Status of Binder Jetting Technology

Metal binder jetting is promising to lower manufacturing cost and lead time for complex geometry and design compared to the conventional manufacturing method [11]. One of the most common alloys in numerous industries is SS 316L which has been studied by many researchers [27]. However, breakthroughs in BJ of SS 316L has not been summarized and reviewed yet to the best knowledge of authors. The aim of this article is to provide guidance for conducting experiments and improving parameters/set-up to build functional 316L parts with competitive properties to wrought or pressed and sintered SS 316L parts. To achieve this, the current status of published studies is reviewed and summarized.

The current status of binder jetting of SS 316L provides an extensive data on selection of feedstock (e.g., powders) and sintering profiles to obtain near full density. However, there is a gap in the current literature for selecting the optimal binder and identifying the role of binder on the final density and dimensional accuracy of the parts. Furthermore, mechanical properties in current literature need to be evaluated and analyzed extensively to report reliable and repeatable data.

## 3. Binder Jetting Applications

Development of AM technology has moved its application from rapid prototyping for design verification to building functional parts and tooling [15]. 3D printing was the first technology that fabricated ceramic molds (for investment casting of metal) directly from CAD model with no extra steps. 3D printing of ceramic shell with integral cores has remarkably enhanced the casting process of metals as compared to the traditional lost-wax casting process [33], as shown in Figure 2.

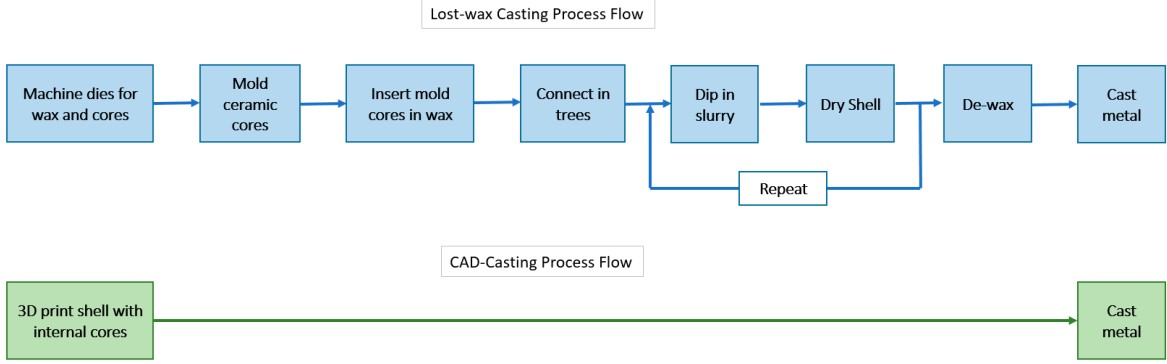

**Figure 2.** Process flow for lost-wax casting and computer-aided design (CAD)-casting [33].

The complex geometry of the structural ceramic components, as shown in Figure 3, can be fabricated via BJ at a very low cost. BJ-printed structural ceramics parts have potential application in numerous fields, including mechanical design, innovation education, and model production [34].

Direct production of injection molding tooling (Figure 4a), metal components (Figure 4b), and end-use parts are feasible by BJ technology [33].

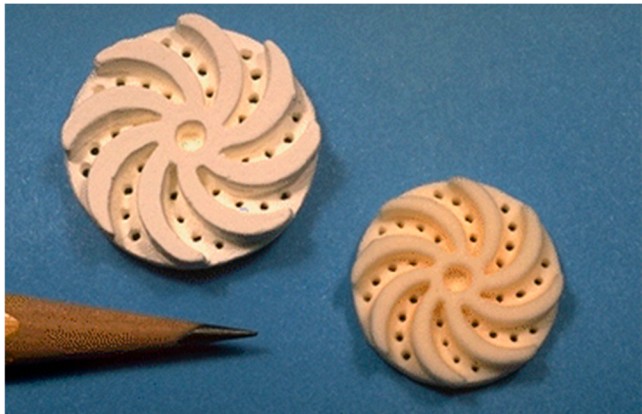

**Figure 3.** Structural alumina part before (left) and after (right) sintering with shrinkage [33].

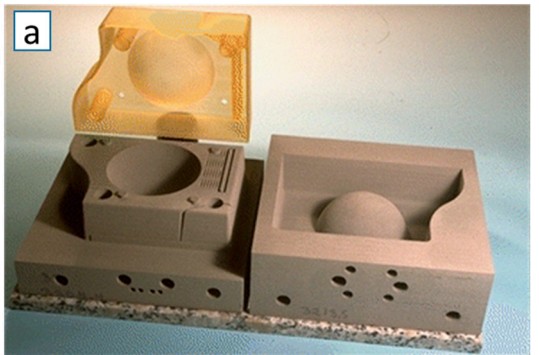
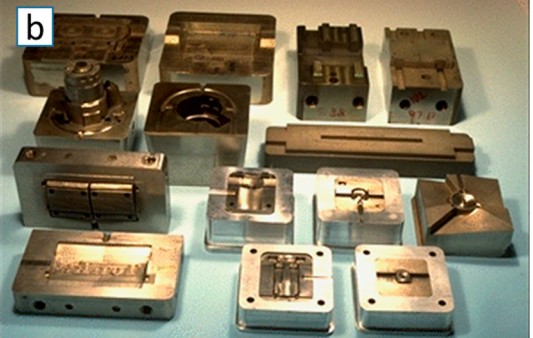

**Figure 4.** (**a**) Injection molding tools and molded part, (**b**) Finished metal tools manufactured by binder jetting (BJ) 3D printing [33].

3D printing has medical and biomedical applications for the production of time-release drug-delivery devices [33] and bone substitute implants [34]. Fabrication of metal matrix composites (MMCs) parts can significantly benefit from BJ technology. While MMCs have not been widely adopted in many applications due to the high cost of manufacturing and geometrical limitations, BJ enables low-cost and scalable manufacturing of MMCs with minimal geometric constraints [35,36]. For example, TiC/Ti–Cu composites have been synthesized by pressureless reactive melt infiltration of a TiCu alloy into porous carbon preforms fabricated by 3D printing [37]. Furthermore, BJ has been used to fabricate porous ceramic filters with a complex internal structure and higher efficiency than conventional filters. One application of this filter is to remove particles from stack gases in coal-burning power plants. Another application of BJ technique is building concept models for rapid prototyping [33].

Metallic parts produced via BJ can be used as functional prototypes or for small-scale production. Some BJ machines which use silica or foundry sand as the powder can produce molds and cores for sand casting applications. BJ also has a sizable application in the automotive industry and heavy equipment industries [26]. MIT has licensed the 3D printing technology to several companies for various fields of use. Licensees and their field of activities are listed in Table 1.

**Table 1.** Licensees by MIT for 3D printing technology.

| Companies | Field of Using 3D Printing Technology |
|---|---|
| ExOne | Metal parts and tooling [33] |
| Voxeljet | Investment casting [38] |
| Soligen Inc. | Ceramic molds for metal casting [33] |
| Specific Surface Corporation (CeraNova) | Production of porous ceramic filters [33] |
| TDK Corporation | Production of ceramic components for electronic devices [33] |
| Integra LifeSciences Holdings Corporation (formerly Therics Inc.) | Production of time-release drug-delivery devices [33] |
| 3D Systems (formerly Z Corporation) | Rapid prototyping and concept modeling [33] |

Binder jetting technology can significantly reduce the time to market for new products (Figure 2), decrease the product cost by limiting the tooling, and improve product quality by enhancing the design. Furthermore, flexible processes generate novel solutions to engineering problems [33]. For instance, Scheithauer et al. [39] introduced a thermoplastic 3D printing approach to fabricate metal-ceramic composites using a high-filled iron-chromium alloy of 17-4 PH suspension and zirconia suspension in molten thermoplastic binder systems with powder content of 50 wt.% steel and 50 wt.% zirconia. This process can be adopted for any materials compositions, which results in manufacturing enhanced quality parts with no delamination or structural changes [39–41].

## 4. Materials

A wide range of materials can be manufactured using BJ technology, including polymers, ceramics, and metals. Despite many efforts to produce metallic parts using BJ technology and evaluate mechanical performance, current studies on BJ of metals have not been reviewed and discussed in any review paper. This work, for the first-time, reviews printing and post-processing parameters in BJ of metallic components, in particular, stainless steel (SS) 316L parts. Because it is the most common engineering steel with numerous industrial applications and thus, is the most studied alloy in BJ process. This review article summarizes composition, properties at low and high temperature in BJ manufactured SS 316L, and provides a comparison with the conventionally manufactured SS 316L parts.

### 4.1. Stainless Steel 316L Composition and Properties

Principal components, in addition to iron, are Cr (15–26 wt.%) to improve corrosion resistance, and Ni (5–37 wt.%) to stabilize the austenite phase. Low-carbon-content in austenitic stainless steels such as 316L, 304L, and 317L can minimize intergranular fracture. The microstructure matrix of Austenitic stainless steels is a solid solution with a high work hardening exponent and low stacking fault energy. A various number of phases can present in the microstructure of austenitic stainless steels, including carbides and intermetallic phases [42,43]. Stainless steel 316L is austenitic stainless steel, frequently used in a wide range of industrial applications because of its high strength and corrosion resistance [44,45], oxidation resistance [46], high heat resistance, and good weldability [45]. Table 2 shows the composition of the cast 316L and 316L P/M.

**Table 2.** 316L stainless steel (SS) composition (wt.%) [47].

| Alloys | Elements | | | | | | | | |
|---|---|---|---|---|---|---|---|---|---|
| | C | Mn | Si | Cr | P | Ni | Mo | S | Fe |
| Cast SS 316L | 0.011 | 0.14 | 0.75 | 17.1 | 0.007 | 12.9 | 2.33 | 0.006 | Balance |
| SS 316L P/M | 0.023 | 0.18 | 0.82 | 16.3 | 0.018 | 13.8 | 2.15 | 0.007 | Balance |

### 4.2. Common Defects of SS 316L Components at Elevated Temperatures

At high-temperature applications, SS 316L components can be susceptible to oxidation, corrosion, carburization, sensitization, and hydrogen embrittlement. Stainless steels are known for high corrosion resistance due to the formation of chromium oxide ($Cr_2O_3$) passive film on the surface of the material that protects the surface from further oxidation [48], and their oxidation resistance increases with chromium content [42]. Austenitic stainless steels usually demonstrate good resistance to general corrosion in different aggressive aqueous environments; however, can opt to localized corrosion such as stress corrosion cracking or pitting [49] at temperatures higher than 400 °C. Furthermore, sensitization could occur during heat treatment or welding processes at 500–800 °C. Sensitization occurs due to precipitation of chromium carbide such as $M_{23}C_6$ along the grain boundaries (at 538–816 °C) leading to chromium depletion at grain boundaries [50,51]. Once the content of Cr at depleted zones falls <12–13 wt.%, the passive film can be easily broken in aggressive solutions causing intergranular corrosion [51]. However, susceptibility to intergranular corrosion decreases in austenitic stainless steels such as 316L, when reducing the carbon content to <0.03 wt.% [50]. Diffusion of atomic hydrogen into metals known as hydrogen embrittlement (HE) can degrade their mechanical properties, including tensile and fatigue properties. HE can take place either at low temperature due to the high concentration gradient or at high temperature due to increased solubility. However, lower diffusivity of hydrogen in FCC structure, as opposed to BCC structure makes austenitic stainless steels less susceptible to HE [52,53].

The amount of porosity of powder metallurgy (P/M) in stainless steels significantly affects their oxidation behavior and makes them more susceptible to the oxidation than conventional stainless steels. One practical approach to reducing the oxidation rate of austenitic stainless steels is the addition of reactive elements. It has been known for decades that using a reactive element like yttrium can improve oxidation resistance of wrought austenitic stainless steels. Similarly, the effectiveness of yttrium addition on the oxidation resistance of P/M austenitic stainless steels has been reported [46].

### 4.3. Additively Manufactured SS 316L Parts

Stainless steel 316L parts fabricated by AM, consist of γ-austenite and δ-ferrite, in contrast, to fully austenitic phase when processed by conventional manufacturing methods. Generally, AM austenitic stainless steels exhibit higher yield strength, ultimate tensile strength and hardness, and lower ductility compared to conventionally processed components [10,54]. Stainless steel 316L components can be manufactured using several AM techniques. Selection of the appropriate method highly depends on the application. Considering manufacturing SS 316L parts using selective laser melting (SLM) and BJ technologies, SLM 316L would offer better mechanical properties and less porosity compared to BJ one. However, if the 316L part is operating in electrochemical or biomedical applications, BJ method would be the better option since internal porosity may be advantageous and strength is not a sole decisive factor [5].

Sintered stainless steel has lower corrosion resistance compared to either cast or wrought stainless steel due to a crevice corrosion mechanism taking place at the pores [47]. Maintaining the corrosion resistance of steel is one of the most critical factors in sintering profile. This can be done by reducing porosity and preventing Cr loss. 316L can obtain near full density if sintering profile is appropriately adopted [27,55,56]. For example, loss of Cr has to be prohibited through applying partial pressure of Ar in vacuum sintering. Figure 5 shows a typical porosity and microstructure of BJ 316L and sintered at 1360–1380 °C. High sintering temperature introduces a higher level of delta-ferrite phase, and there may even remain some ferrite residues after cooling. It is generally known that up to 8% of delta-ferrite has a negligible effect on mechanical properties and corrosion behavior of steels [57]. Higher sintering temperature could lead to higher density at a cost of dimensional accuracy loss [4,55]. 316L is typically expected to obtain 96 ± 1% final density with acceptable dimensional accuracy [27].

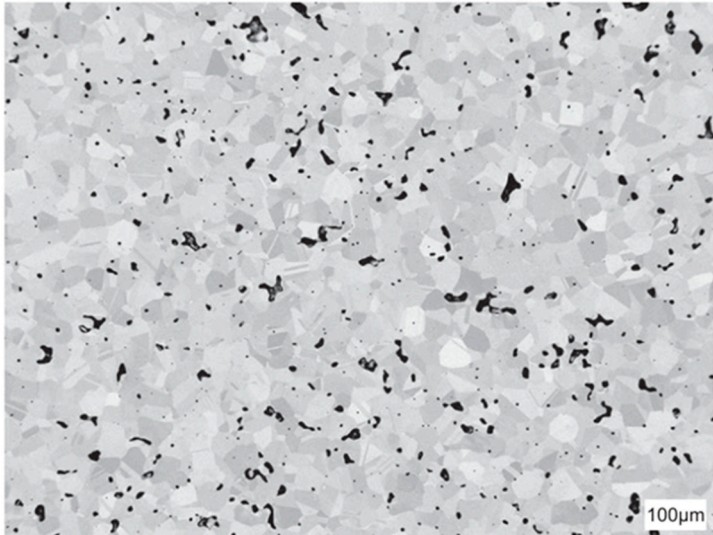

**Figure 5.** Porosity and microstructure of sintered SS 316L at 1380 °C in vacuum atmosphere [27].

Several parameters and variables affect the final part properties including design features, feedstock properties; powder and binder properties, printing parameters (such as layer thickness, printing saturation, heater power ratio, and drying time), and post-processing parameters (including sintering temperature, time, environment, and heating/cooling rate). Feedstock properties, printing parameters, and post-processing properties are reviewed and summarized in the following subsections.

## 5. Binder Jetting Feedstock Properties

### 5.1. Powders

Powder properties influence processability and quality of BJ manufactured part [58]. Physical properties of the powders, including flowability and bulk density, mainly determine final density. However, the mechanical properties of parts are affected by powder shape and particle size distribution (PSD) [34].

Flowability of the powders influences the spreadability. Powders without a good flowability, will not be spread smoothly in the bed and lead to defective part structure [28,34]. In powder bed AM technologies including BJ, powders are deposited in the bed without any external forces (e.g., extrusion); thus the bulk density of powders affects the green density of the part. Green density subsequently determines the sinterability of the powder compact [34].

Size of the powder in BJ is in the range of 0.2–200 μm [59]. Powder size significantly affects flowability, the reactivity of the powder with the binder, wettability with the binder, surface roughness, and resolution of the part [60,61]. Figure 6 shows the micrographs of a variety of metallic powder particle size from fine to large used in BJ [31]. Fine powders have poor flowability compared to coarse particles because interparticle friction inhibits particles from sliding and packing well. Thus, small particles with large surface area and high interparticle friction due to large van der Waals forces, have less flow and poor packing properties [28,62,63].

It has been reported that flowability improved with increasing median particle size [60]. Finer particles have potentially better sinterability, thinner layers, and higher surface area per unit volume, whereas larger particles have better spreadability and lower surface area per volume [28]. The higher specific surface area of fine powders theoretically contributes to better powder-binder reactivity.

Zhou et al. [61] reported that particle size impacts the size and distribution of pores in the bed. Fine powders agglomerate and create large voids inside the powder bed causing slow binder penetration rate, which ultimately leads to low printing resolution [34].

Fine particles have poor wettability due to large contact angle and agglomeration [60]. An investigation on the effect of particle size on surface roughness of the part revealed that both coarse and fine powders resulted in rough surface, while a median range of large and small particles achieved lower surface roughness [60].

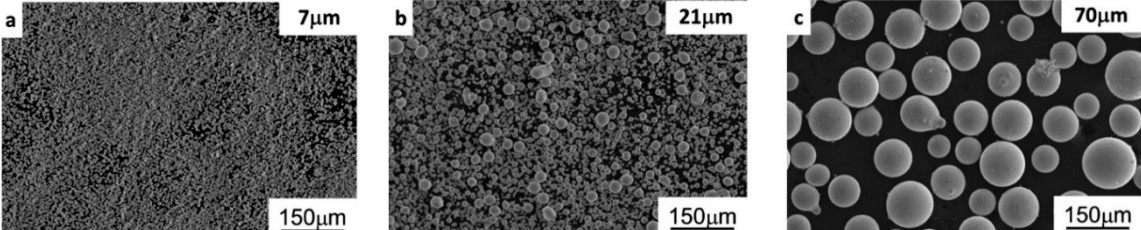

**Figure 6.** Micrographs of gas-atomized Inconel 718; (**a**) 7 μm powders, (**b**) 21 μm powders, and (**c**) 70 μm powders [31].

Particle size distribution (PSD) has a significant impact on BJ process. Bimodal powder size distribution has significantly improved packing density as well as the final density of the part [14]. Because a high amount of fine particles reduces flowability of powder due to high inter-particle cohesiveness [64]. Furthermore, larger particles improve the spreadability of the powder in the powder bed and small particles fill the pores and interstices between larger particles, thus improve flowability and the overall packing density [28,55]. A study on multi-modal powders reported that high packing density can be achieved with 20–40 Vol.% (volume fraction) of the fine powders [65].

The shape of the powder significantly influences the flowability, tap density, packing density, contact mode between particles, and the porosity of the green part [34]. Metal powders are manufactured via various techniques, including mechanical, chemical, and powder atomization methods that each can result in different shapes of powders [63]. Atomized powders are more commonly used for additive manufacturing (AM) [66]. Although water atomized metallic powders (Figure 7b) are more common and cost-effective in conventional press and sintering P/M, gas atomized powders (Figure 7a) with spherical shape offers better flowability [2] due to lower interparticle friction [50,67,68]. Adding a low weight percentage (up to 2%) of solid lubricant [69] such as zinc stearate [70,71] to feedstock can reduce the interparticle friction, and enhance flowability [30].

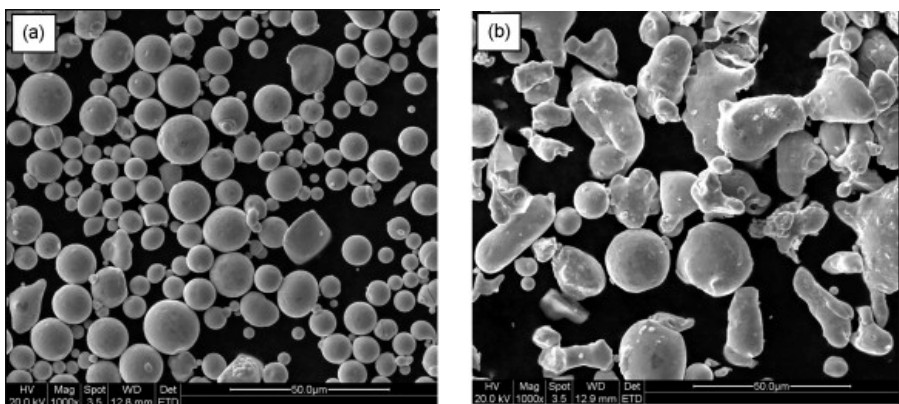

**Figure 7.** SEM images showing characteristic morphologies of 316L SS powder; (**a**) gas atomized powder, (**b**) water atomized powder [72].

Lower tap density may lead to high porosity and consequently higher shrinkage rate and lower final density [27]. Final properties of the part can be affected by adding fibers. The fibers can be polymers, ceramics, fiberglass, and graphite [73,74]. While larger fibers up to the size of the layer thickness reinforce the final part, smaller fibers up to the half-size of layer thickness may

increase dimensional stability. However, adding too many fibers increase the interparticle friction and subsequently lowers the spreadability of the powder in the bed [28].

Powder deposition method depends on powder flow characteristics. The ideal powder flows freely with negligible inter-particle forces. Generally, larger particles (>50 µm) have good flowability; however, with particle size reduction, larger forces act between particles causing lower flowability. [75]. The powder can be deposited in both dry and slurry-based methods. Dry deposition is more favorable due to the simplicity, speed, and ease of testing. Coarser particles (>20 µm) are preferred to be deposited in the dry state. Dry deposition uses a counter-rotating roller to spread the powder. Roller deposits a new layer of powder without disturbing the previous layers [28]. The roller can be vibrated to facilitate spreading [76].

Finer powders (<5 µm) with larger electrostatic forces than gravitational forces have low spreadability that causes challenges for dry powder deposition [77]. A solution to overcome the difficulties is to disperse fine powders in a liquid to form a slurry. In the wet or slurry-based deposition, each layer is formed by rastering a jet of ceramic slurry (typically about 30 vol. % ceramic in water or water-alcohol) over a porous substrate [76].

### 5.2. Binders

Binder is simply a temporary glue to join the powder particles into the desired shape and holding the particle in that shape until initial stage of sintering [78]. One of the most common types of binders is an organic liquid binder. Organic binders, including butyral resins, polymeric resins, and polyvinyls leave a slight residue when thermally decomposed and can be mixed with a variety of powder materials. While the printing process is similar for all materials, binder selection could improve the green part properties [19,27]. Maintaining the liquid phase of the binder is essential because the dried liquid in the printhead can cause clogging in the head [55].

All cured binder should be evaporated during debinding at the beginning of sintering. Most organic binders burn off at 200–300 °C [79]. Any residual binder in the compact will turn into carbon and may later diffuse to surface of the part at high sintering temperature leading to carburization [28,55,80].

Liquid Binder can be applied in BJ process using two conventional ink-jet printing technologies; continuous-jet (CJ) and drop-on-demand (DOD) [28,79,81]. In the CJ system, pressurized binder passes narrow orifices to form columns of liquid jets [79]. In DOD technique, droplets are formed when required for printing by applying electrical impulses. Piezoelectric and thermal inkjet heads are typical DOD printheads [82]. Piezoelectric technology works based on electrical stimulation of a piezoelectric material which causes physical deformation of piezo material [81]. Piezoelectric heads eject binder droplets when the piezoelectric transducer generates a pressure wave on the binder reservoir [79], while thermal inkjet heads vaporize the liquid in the printhead and benefit from volume expansion to eject the ink from the printhead [28].

The vaporized liquid in the thermal printheads should dissolve quickly to avoid deposition of solid material in the printhead [28]. The CJ method is more beneficial for high-speed coverage of relatively large areas, while DOD technology is more applicable for depositing small and a controlled amount of binders [79]. Among DOD systems, piezoelectric printheads may offer greater reliability and more potential for the development of a wide range of binder composition, due to utilizing mechanical action rather than thermal one [81].

## 6. Binder Jetting Printing Parameters

### 6.1. Layer Thickness

Layer thickness is an important process parameter for several types of AM techniques since it defines the smallest possible design feature size and constraint resolution. Layer thickness is equal to the distance by which build plate is lowered after printing a layer. The capability of BJ machine determines the layer thickness variation range. Powder size and characteristic may partially determine

the desired layer thickness [83]. While thinner layer thickness results in better final part properties, building time consequently increases [15].

The different layer thickness of 3× of particle size [62,84], 2× of particle size [85], and >the largest particle [31,86,87] has been suggested to maintain good flow and spreadability. During 3D printing, binder penetrates vertically and laterally into the interstitial sites of the particles [85]. For instance, when selecting too thin layer thickness such as 20 μm for the powder of all particles <20 μm and mean particle size of 5.5 μm [85], penetration of excessive binder to unwanted sites would result in poor resolution [88,89]. However, too thick layer thickness such as 50 μm [85], could not provide sufficient time for the binder to penetrate to printed layer in a vertical direction [85,88,89]. With an optimal layer thickness of 35 μm, binder penetration in vertical and lateral directions could proceed to provide the desired strength for the printed part [85].

*6.2. Printing Saturation*

During printing, the build bed is filled with powder, pores or air, and binder. The percentage of the air volume, which is occupied by the binder, is called printing saturation or binder saturation and can be defined by the following equation [15]:

$$S = \frac{V_{binder}}{V_{air}} = \frac{V_{binder}}{(1 - PR) \times V_{solid}} \tag{1}$$

where "*PR*" is packing rate, which is the fraction of powder volume in a defined volume of powder and air. Packing rate is defined by the following equation [15]:

$$PR = \frac{V_{powder}}{V_{powder} + V_{air}} \tag{2}$$

Packing rate is usually 50–70%. Saturation describes the amount of deposited binder. Therefore, saturation implies printing stability. Low saturation may fail due to lack of enough binder to firmly join the powder particles together whereas high saturation rate may affect the geometry of the part due to binding excessive amount of powder. Figure 8 illustrates a schematic of low and high printing saturation, respectively [15].

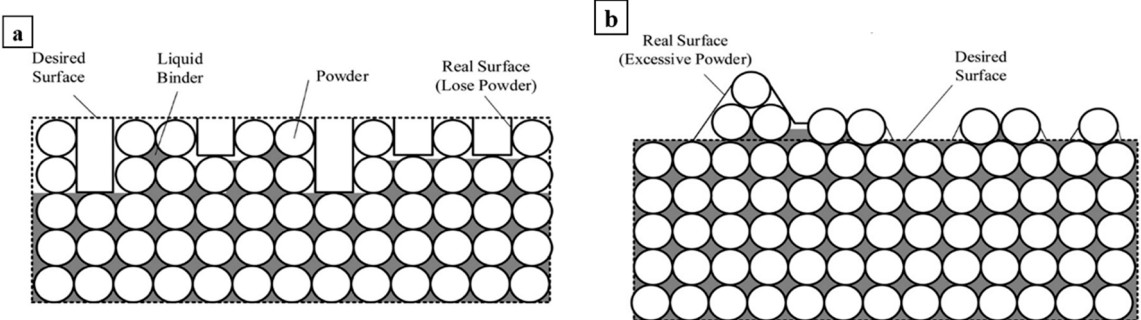

**Figure 8.** Schematic of printing saturation; (**a**) low printing saturation results in lack of enough binding, (**b**) high saturation results in excessive powder bond [15].

*6.3. Heater Power Ratio*

After printing of each layer, the build bed will be moved under a heater to dry the binder. The heater power ratio indicates the ratio of current heater power to the maximum heater power to determine the heating speed and temperature. Required heater power varies in different printing settings [30] with no specific limit that can be defined. Too low power ratio cannot dry the binder, while too high power ratio consumes more power energy and may increase deformation and shrinkage rate during printing [15]. Therefore, the heater power ratio needs to be set not to cause any defects

in the part. Heating power of 75% with binder saturation of 70% has been used for printing 316L on Exone M-Lab printer [90].

### 6.4. Drying Time

Drying time is then the duration of drying the binder under the heater after printing each layer. During drying, printhead moves into the cleaner tank to remove excessive binder to prevent the printhead from blockage. Thus, short drying time results in printhead blockage, which has a significant impact on final part surface quality [15]. Figure 9 displays a schematic of short drying time when binder partially penetrates to deposited powder and leaves more pores in the printed part.

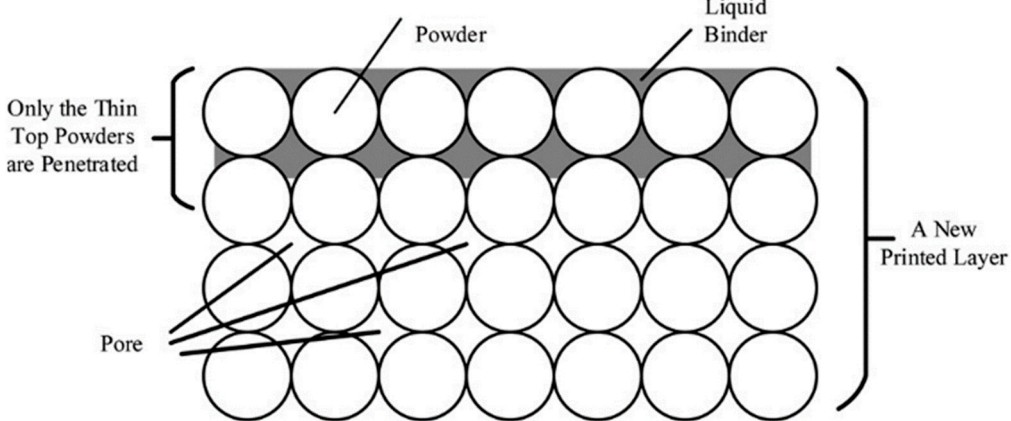

**Figure 9.** Short drying time results in inadequate binder penetration [15].

## 7. Binder Jetting Post-processing

### 7.1. Binder Curing

While printing, the powder bed is undergone interlayer drying using an overhead heater [91]. When printing all of the layers is completed, the 3D part is still embedded in a loose powder. Then, post-print curing is used to further polymerize (cure) the organic binder, dry the solvent crosslink polymers [91], and improve the structural integrity of the component when still surrounded by loose powder [55]. Curing can be conducted using visible light, heat, or pressure [28].

Bai et al. [91] printed copper using a metal organic deposition method which deposits metal nanoparticle ink, and can be used as binder precursor as a replacement for polymeric binder. The microstructure of part prior and after curing is investigated by Bai et al. [91]. Before curing, no evidence of nanoparticle sintering was found, as shown in Figure 10a,b. However, after curing, sintered nanoparticles were observed at the necking of powder particles, as shown in Figure 10c,d. It was concluded that metal nanoparticle ink built less strong green part compared to polymeric binders. However, the part could maintain structural integrity after de-powdering with compressed air [91].

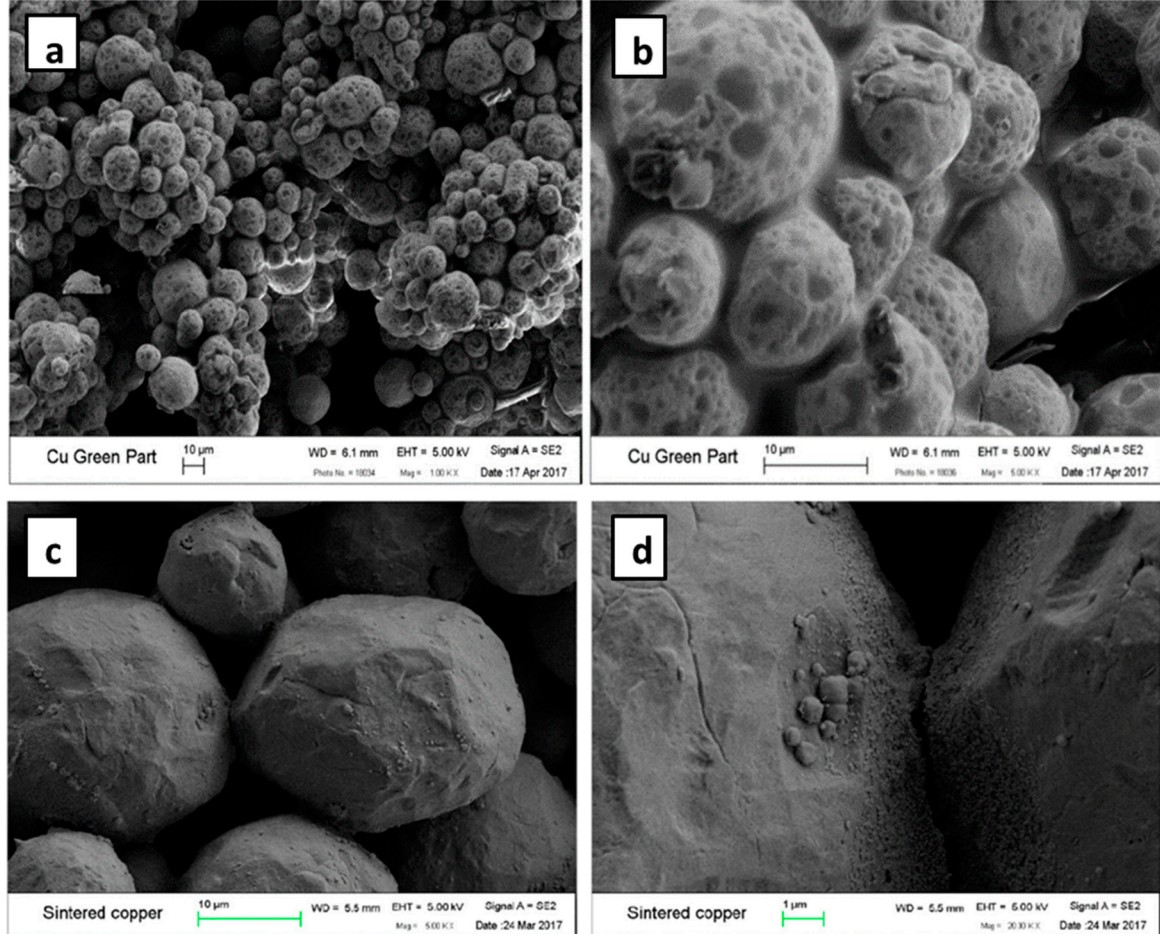

**Figure 10.** Micrographs of copper parts printed with metal-organic-deposition ink (150% binder saturation (**a**) green and without curing at magnification of 1kx, (**b**) green and without curing at magnification of 5kx, (**c**) bonding formed by sintering of copper nanoparticles at magnification of 5kx (**d**) sintered at magnification of 20kx [91].

## 7.2. Depowdering

After curing, the loose powder has to be removed from the green part. Brushing away the excess powder is an effective depowdering method for parts with no internal features. Complex geometries with internal features might require more careful steps. Other dry depowdering methods include blowing air, vibration, and vacuuming. When the binder is not soluble in the fluid, wet depowdering including ultrasonicating, microwave-induced boiling, and $CO_2$ bubble generation in soda water are used as well [28,91].

## 7.3. Debinding

Debinding occurs during the initial stage of sintering where binder becomes unstable and decomposes into constituents that can evaporate. Conventional polymers contain the same basic carbon–carbon, carbon–oxygen, and carbon–hydrogen bonds which burn out over the same temperature ranges. Most of the binders melt at temperatures <150 °C but do not evaporate until 300–500 °C. Adding an active agent, such as oxygen can accelerate the burn-off process [63]. Debinding is an essential step; otherwise, the residual carbon content of binder may diffuse inside the part during sintering and can negatively affect part ductility. To burn-off the binder, depowdered parts need to be placed in a furnace that can be the same furnace for sintering. For 316L parts, most of the binder phase usually burns off at approximately 450 °C [55].

### 7.4. Sintering

Sintering is contacting the particles to bond at high temperature that occurs by solid-state diffusion at a temperature below the melting point [63]. However, sintering may involve liquid phase formation [92,93] known as liquid phase sintering. Liquid phase sintering rather than solid state sintering offers better control on microstructure, less porosity in the sintered parts, and faster sintering at lower sintering temperature [94].

Driving force of sintering is the reduction of the total interfacial energy of the powder compact that can be expressed in the following equation [93]:

$$\Delta(\gamma A) = \Delta\gamma A + \gamma\Delta A \tag{3}$$

where $\gamma$ is the specific surface energy, and $A$ is the total surface area of the compact. Interfacial energy change ($\Delta\gamma$) is attributed to densification which is the replacement of solid/vapor surface by solid/solid surface in solid-state sintering, and surface area change ($\Delta A$) is due to grain coarsening (grain growth) [93].

At melting temperature, materials may exhibit a very high rate of atomic jumping whereas the rate of atomic jumping during sintering is only 1% of that for melting temperature. Thus, small fracture of atomic jumps leads to sintering bond growth and surface energy reduction of small particles. Smaller particles sinter faster because they require fewer atoms to fill in the neck due to shorter moving distance and larger stress. Bonding starts as neck grows between adjacent particles and sintering continues to progress as bonds enlarge, merge and grain boundaries grow to replace the solid–vapor interfaces. Sintering is finally accomplished by pores elimination and microstructure coarsening [63].

To sinter parts with reproducible properties, sintering variables, including sintered density, grain size, and pore distribution should be controlled. Sintering variables related to powder feedstock including powder chemistry, particle shape, size, and particle size distribution can affect grain growth and densification. Another group of sintering variables is process variables including sintering temperature and time, heating and cooling rate, holding time, pressure, and sintering atmosphere [78]. Material variables affect the compressibility and sinterability of the compact while process variables mostly include thermodynamic variables [93].

Solid state sintering has three overlapping stages, including initial, intermediate, and final stage [63,93]. The initial stage is determined by the formation of necks between particles [93] and involves irregular and angular pores [63]. Only 2–3% of densification occurs at the initial stage.

The intermediate stage includes open pores. Curvatures around pores continue transferring mass to fill in of concave regions. At the intermediate stage, necks are large enough to interfere and overlap. Grains grow and pores shrink mainly through particle re-arrangement [63,95]. Up to 93% of the relative density occurs at the intermediate stage before pores isolation [93].

The final stage involves mostly closed, and spherical pores [63] and densification occurs from isolated pore state to the final densification. After full densification, grain growth accelerates since no pore exist to impede the process [63]. Channel pore model and isolated pore model are typically used for shape changes of pores during the intermediate and final stage sintering, respectively [93].

Considering the same size spherical particles, sintering of powder compact can be represented by the two-particle model [63,93]. Figure 11 displays necking between two spherical particles where $x$ is neck radius, $a$ is particle radius, $\theta$ is the dihedral angle between particles, $r$ is the radius of neck curvature, and ($x/a$) is neck-size ratio [93].

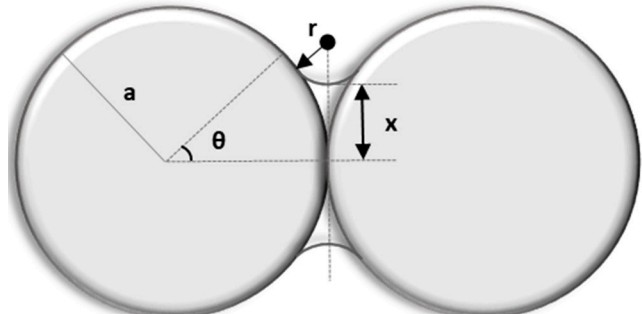

**Figure 11.** Sintering of two spherical particles for the initial stage of sintering with a neck radius of '*x*' and particle radius of '*a*' with a neck curvature with a radius of '*r*.'

The differences in bulk pressure, vacancy concentration and vapor pressure determine sintering driving force [93]. Figure 12 demonstrates the mechanisms of material transport as follows [96]:

- Lattice (volume) diffusion (from grain boundary)
- Grain boundary diffusion (from grain boundary)
- Viscous (plastic) flow (from bulk grain)
- Surface diffusion (from grain surface)
- Lattice diffusion (from grain surface)
- Gas phase transport (from grain surface):

  ○ Evaporation/Condensation
  ○ Gas diffusion

Mechanism:
1. Surface diffusion
2. Lattice diffusion (from the surface)
3. Vapor transport
4. Grain boundary diffusion
5. Lattice diffusion (from grain boundary)
6. Plastic flow (by dislocation motion)

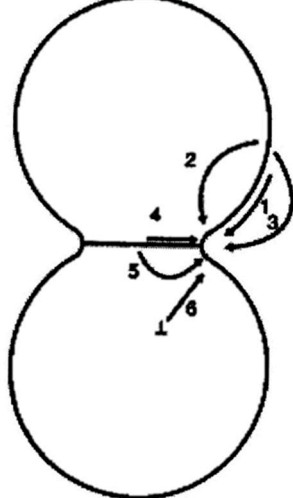

**Figure 12.** Material transport paths during sintering [96].

Not all of the material transport mechanisms contribute to shrinkage and densification. The densifying mechanisms are lattice (volume) diffusion, grain boundary diffusion from the grain boundary to the neck, and plastic flow causing neck growth and densification. Non-densifying mechanisms, including vapor transport, surface diffusion, and volume diffusion from the surface of particles to the neck result in neck growth and particle coarsening without any densification. If non-densifying mechanisms are dominant mechanisms a porous microstructure will be obtained [93,96].

Diffusion is an essential sintering mechanism [93]. Diffusion is a temperature dependent rate process, and diffusion coefficient ($D$) is [97]:

$$D = D_o \exp\left(-\frac{Q}{RT}\right) \tag{4}$$

where $D_o$ is a constant with the same unit as $D$ (cm$^2$/s), $Q$ is the activation energy (J/mol), $R$ is universal gas constant (8.314 J/mol·K), and $T$ is absolute temperature (K).

The lattice diffusion of atoms from grain boundary to the neck enables boundary to act as a vacancy annihilation site. During lattice diffusion of atoms from grain boundary to the neck surface, the neck region is under tensile stress and grain boundary is under compressive stress. By this sintering mechanism, the material is removed from the particles contact area, which can result in neck growth and densification (shrinkage) [93]. This mechanism is similar to that take places in Nabarro-Herring creep [98]. Nabarro-Herring (NH) creep occurs as a result of diffusion of vacancies from high chemical potential ($\mu$) regions at grain boundary subjected to normal tensile stresses to lower chemical potential regions subjected to normal pressure. This produces a normal velocity of grain boundaries or a strain rate [98,99].

Grain boundary (GB) diffusion involves material transport from the grain boundary to the neck, which is similar to that occurring during Coble creep [93]. Coble diffusional creep occurs through the diffusion of atoms in a material along grain boundaries [100].

Herring's scaling law [101] explain the effect of particle size on sintering kinetics. For powders of similar shape in the same experimental condition and sintering mechanism, scaling law predicts the relative duration of sintering time to get the same degree of sintering. For a smaller powder with $a_1$ radius and larger powder with $a_2$, where $a_2 = \lambda a_1$ [93];

$$t_2 = \lambda^\alpha t_1 \tag{5}$$

where $\lambda$ is constant, and $\alpha$ is an exponent. The general form of sintering equation can be expressed as [93]:

$$\left(\frac{x}{a}\right)^n = F(T)a^{m-n}t \tag{6}$$

where $x$ is neck radius, $a$ is particle radius, $F(T)$ is a temperature-dependent function, $t$ is required sintering time, $m$ is an exponent of grain size for densification, $n$ is grain size exponent for grain growth and $a^{m-n}t$ should be constant, thus $\alpha = n - m$. Similar shape assumption of scaling law is not generally satisfied in sintering mechanism since grain grows. Nevertheless, it provides a simple way to demonstrate the impact of particle size on microstructural changes [93,102].

Coble's model for intermediate stage sintering is based on BCC packed tetrakaidecahedral grains with cylindrical pores along all of the grain edges. Intermediate stage Coble's microstructure model assumes equal shrinkage of the pore in a radial direction. Lattice diffusion and grain boundary diffusion are two available mechanisms at this stage [34,93]. For the final stage sintering, Coble's model assumes tetrakaidecahedral grains with spherical pores and concentric diffusion of atoms to the surface of the pore [93]. At the final stage, the densification rate is inversely proportional to the grain size. Coble's model does not consider grain boundary as the atom source for densification. Thus, for the final stage, Herring's scaling law [103] can be used to take into account the impact of pore surface area on the material flux from grain boundary to the pore [93]. Densification at the final stage is interrelated with grain growth in the presence of pores [93,96]. It has been reported that the densification rate decreases as particle size and sintering time increase. However, the grain boundary diffusion mechanism dominates for the finer powder [93].

In BJ sintering processes, starting with generally a 60% green density, the initial stage of sintering leads to 1.5% to 2% shrinkage as a result of necking between particles. However, the final stage of sintering may result in 15% linear shrinkage. A mixture of metals with different liquids and distribution of large and fine particles can profoundly impact the sintering quality and densification behavior

of the component [28]. Metal powders typically need mean particle size $D_{50} < 20$ μm to obtain full density [104]. Multi-modal powders usually achieve higher green density; however, not all of them result in higher densification. Typically for BJ processes most of the near full density (>90%) was achieved via liquid phase sintering that got promoted when sintering additives were utilized [75]. Additional post-processing can be performed to obtain full density, including hot isostatic pressing (HIP) and infiltration.

*7.5. Infiltration*

To achieve near full density infiltration could be used if compositional changes do not affect part performance, and introduction of a new element does not deteriorate mechanical properties of the part. Low and high-temperature infiltrations are possible options depending on the part material and binding mechanism. The infiltrant must be melted at a temperature < melting point or solidus temperature of the bulk material to prevent the part from losing structural integrity [28]. For the stainless steel materials, a low melting material such as bronze is used as an infiltrant which can fill the open pores of the printed powder to improve the final density [4,26]. Cordero et al. [105] manufactured ferrous powder via BJ using molten bronze as infiltrant and reported that infiltration could improve the strength of the sintered component by eliminating the stress concentration points at interparticle necks as shown in Figure 13 [105].

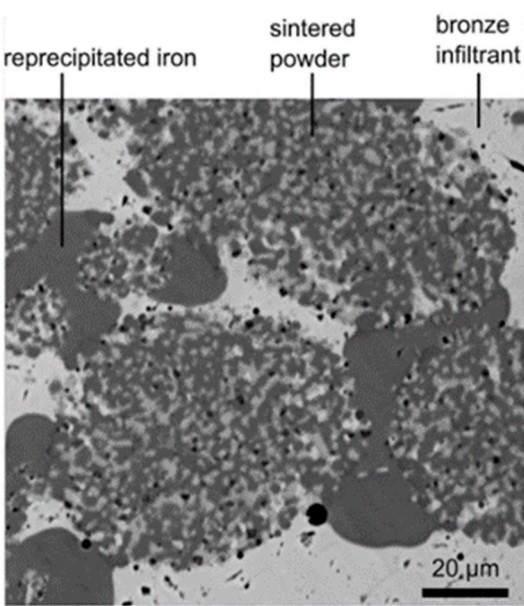

**Figure 13.** Backscatter electron micrograph of the infiltrated ferrous material with bronze infiltrant [105].

## 8. Review of Previous Studies on Binder Jetting of SS 316L

Influence of powder size and shape, sintering profile including time and temperature, sintering additives, sintering atmosphere, and additional heat treatment after sintering (HIP) on the final density, surface roughness, mechanical properties, and dimensional accuracy of BJ manufactured SS 316L parts have been investigated by many researchers [4,14,27,55,56,78,106,107]. However, these studies [4,14,27,55,56,78,106,107] have not been reviewed, summarized and compared in a review article. Table 3 provides an overview of BJ manufactured SS316L. Reported results in the literature are summarized in the following sections.

**Table 3.** Summary of BJ SS316L literature and factors have been investigated.

| Parameters | Impact on Final Density | References | Impact on Shrinkage Rate | References | Impact on Mechanica [1] Properties | References |
|---|---|---|---|---|---|---|
| Powder size and shape | √ | [4,55,56,78,106] | √ | [55,106] | √ | [56] |
| Sintering profile (including temperature and time, and ramp-up) | √ | [56,78] | √ | [14,108] | √ | [55,56] |
| Sintering additives | √ | [55] | √ | [55] | √ | [55] |
| Post-processing (hot isostatic pressing (HIP)) | √ | [27] | × [1] | | × | |
| Binder | × | | × | | × | |
| Printing parameters (Orientation, layer thickness) | × | | × | | √ | [107] |

[1] This sign indicates that few or none studies have been reported on the specific area.

### 8.1. Role of Powder Size and Shape on Binder Jet 316L Parts

Rego et al. [55] investigated the impact of particle size and mixture ratio on green and sintered density of SS 316L parts using four different sizes of SS 316L powders. The powders were labeled as S (small), M (medium), D (default size, 30 μm), and L (large) with the mean particle size of 4, 14, 30, and 82 μm, respectively as summarized in Table 4. Powder mixtures with two and three different particle sizes were prepared. Samples with dimensions of 8 mm × 8 mm × 8 mm were printed using X1-lab by ExOne and mixture of ethylene glycol monobutyl ether (EGBE), isopropanol (IPA), and ethylene glycol (EG) as binder [4].

**Table 4.** List of SS316L powder their average particle size.

| Powders | Mean Particle Size (μm) [55] |
|---|---|
| | Large (L) 82 |
| SS 316L | Default (D) 30 |
| | Medium (M) 14 |
| | Small (S) 4 |

The build bed was then placed on a convection oven (DX302C, Yamato, Tokyo, Japan) for curing at 195 °C for two hours. Debinding was performed at 460 °C in an air furnace (KSL-1100X, MTI Corp., Richmond, CA, USA), and the binder was burned out after two hours without noticeable oxidation on the surface. Then, samples were sintered at vacuum furnace (Materials Research Furnaces, Model G3000, Allenstown, NH, USA), at 1200 °C, 1300 °C, and 1350 °C for 6 h to eliminate internal pores before consolidating. Single powder size resulted in 50.77% green density. Among bimodal powders with different mixing ratio, a sample with a ratio of 70D-30S demonstrated 60.3% of green density. Sohn et al. [109] reported that packing density increased if particle size distribution was extended. Gaussian and log-normal size distribution of bimodal powders with 60:40 (large diameter: small diameter) ratio obtained packing density of 78.2% and 77.5%, respectively.

For mixtures with three different particle sizes, the sample with the ratio of 70L-25M-5S resulted in the highest green density of 63.87%. However, these samples attained a final density of 71.6%, 72.7%, and 72.9% at a sintering temperature of 1200 °C, 1300 °C, and 1350 °C, respectively. Previously, McGeary [110] experimented with mixing different sizes of spherical powder. For bimodal mixture with a radius ratio of 7:1 and a mixing ratio of 70:30 [111], the green density of 83% was reported using a vibrating powder bed.

Ziaee et al. [106] investigated the effect of different powder preparation on final density and shrinkage rate of the parts. Three different types of 316L SS powders were used in this experiment. The binder used in this study was an M-Flex solvent-based binder supplied by ExOne, which was added to powders at 90% of the void volume of the powder. Two other feedstocks were powder and nylon mixtures with 25% and 33% nylon volume ratio. Nylon was added to mixtures as a fugitive space holder to increase porosity. The primary purpose of adding nylon was investigating the possibility of fabricating porous structure with the same shrinkage rate as dense parts. Powders, layer thickness, drying time, and binder level used for different powders are summarized in Table 5 [106].

**Table 5.** Powders and printing parameters used in Ziaee et al. [106].

| Powders | Layer Thickness (μm) | Drying Time (s) | Binder Level (%) |
|---|---|---|---|
| 316L < 22 μm | 100 | 12 | 24 |
| 25% Nylon + 316L | 100 | 12 | 24 |
| 33% Nylon + 316L | 100 | 12 | 48 |
| Agglomerates (cured at 100 °C) | 50 | 20 | 48 |
| Agglomerates (cured at 185 °C) | 50 | 20 | 48 |

After printing, samples were cured at 185 °C for four hours. However, the nylon mixes were cured at a lower temperature of 140 °C to avoid bonding with loose powder. Sintering was performed at 95% Ar, 5% $H_2$ atmosphere with 10 SCFH flow rate. Heating rates were 7 °C/min with a 60-min hold at 800 °C to complete decomposition of binder and Nylon. The parts were held at 1360 °C for 60 min. Figure 14 displays the porous structure of all samples after sintering and attaining final density through grain boundary diffusion.

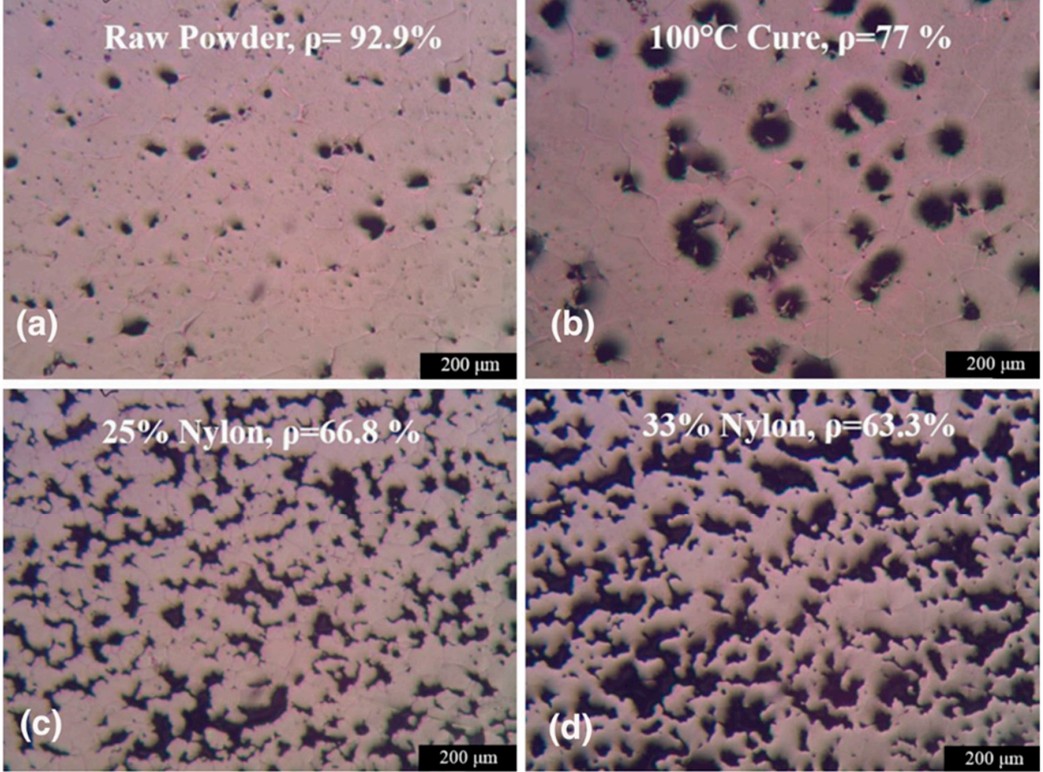

**Figure 14.** Porosity on the cross section of the samples made by feedstock materials: (**a**) Raw 316L < 22 μm, (**b**) agglomerates cured at 100 °C, (**c**) 25% Nylon + 316L, and (**d**) 33% Nylon + 316L [106].

Results of Ziaee et al. [106] revealed that spreadability of powder in the bed had a direct impact on sintered density and sintering shrinkage as shown in Figure 15. Baseline and agglomerate-cured at

185 °C samples with similar spread density, showed a similar sintered density of 92.9% and 93.9%, respectively. However, agglomerate-cured at 100 °C, had a final density of 77% due to lower spread density. For nylon-added samples, density depends on nylon content since nylon decomposes during sintering and leaving pores in the part. Primarily, nylon was added to the mixture as a fugitive space holder to create porous parts. Samples with 25% and 33% nylon, achieved 66.8% and 63.3% sintered density, respectively which makes them suitable for filters, heat exchangers, and various energy applications [106].

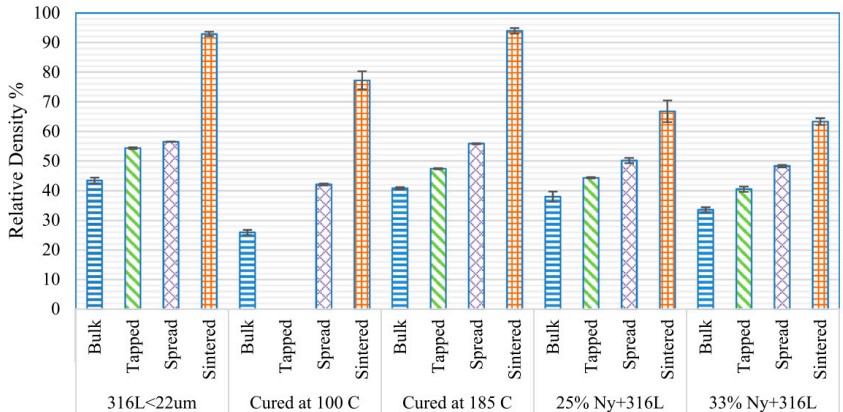

**Figure 15.** Relative bulk, tapped, and spread densities for the raw 316 and processed powders [106].

For all samples, shrinkage was highest in the Z-axis (as shown in Figure 16), suggesting that more porosity exists between layers rather than within layers. Samples with the highest spread density showed the lowest shrinkage rate. However, samples with fugitive agents of 25% nylon showed the same shrinkage rate as agglomerate-cured at 185 °C sample, which had a high final density. It was suggested [106] that since the same steel powder was used for raw and nylon added samples, same sintering behavior was expected for steel region of samples. Therefore, after decomposition and evaporation of nylon particles and leaving pores in structure, the steel region was almost fully dense and revealed the similar shrinkage rate as the raw sample. Thus, it was concluded that there is a possibility of building parts with different final density and consistent shrinkage levels via binder jetting with controlled porosity [106].

**Figure 16.** Dimensional deviation for X, Y, and Z directions for the sintered samples [106].

Verlee et al. [56] investigated the effect of particle size, particle shape, sintering time, and temperature on final part density and mechanical properties. Four spherical 316L stainless steel powders were used: 22 and 31 μm ($D_{85}$) powders from Carpenter, and 20–53 and 45–90 μm from Hoganas. Two non-spherical powders of 45 μm and 45–90 μm from Hoganas were used. List of powders is given in Table 6.

**Table 6.** Powder density and particle mean diameter [56].

| Powders | Apparent Density (g/cm$^3$) | Particle Mean Diameter (μm) |
|---|---|---|
| 22 μm | 4.69 | 13 |
| 31 μm | 4.58 | 17 |
| 45 μm Ø | 2.77 | 27 |
| 20–53 μm | 4.38 | 38 |
| 45–90 μm Ø | 2.32 | 64 |
| 45–90 μm | 4.44 | 73 |

Prometal RxD printer was used with a build bed equipped with vibrating layer spreading. The layer thickness was set to be 75 μm except for coarse (45–90 μm) powders which were built with a layer thickness of 100 μm. Binder used in this study was Prometal binder PM-B-SR2-02 with a saturation rate of 60% (which means 24% of the total volume of the part is occupied by binder if the packing density of the powder is 60%. Curing was performed in an oven at 180 °C for 2 h. Debinding was performed under a 100% Ar atmosphere with holding at 900 °C temperature for 30 min. Sintering was performed in a separate batch furnace in a hydrogen environment and held for 90 min at different temperatures (varied from 1200 °C to 1430 °C) or at a fixed temperature for different holding times.

Results of Verlee et al. [56] suggested that final density depends on particle size and sintering temperature. For a given temperature, higher density was obtained from smaller particles because the densification rate decreased with increasing particle size. For smaller particles, dominant densification mechanism was grain boundary diffusion. Volume diffusion was the primary bulk transport mechanism for a coarser particle at a much slower rate than GB diffusion [93]. Particle size defines sinterability and final pore size. Table 7 shows densification and porosity percentage for all samples. Smaller particles contain higher surface energy so that higher densification could be obtained. Finer particles also have smaller pores between particles that cause prior densification in the sintering process. Thus, smaller particles can achieve higher density at the lower sintering temperature. Particle shape affects the green density and consequently affects the final open porosity range after sintering. Spherical powders showed a higher final density compared to non-spherical ones [56].

**Table 7.** Pore characteristics of samples sintered at 1395 °C for 90 min [56].

| Powders | Density (%) | Open Porosity (%) | Closed Porosity (%) | Densification ($\psi$) |
|---|---|---|---|---|
| | ±0.1% | ±0.1% | ±0.1% | |
| 31 μm | 97.4 | 0.0 | 2.6 | 0.94 |
| 45 μm Ø | 60.9 | 37.6 | 1.5 | 0.4 |
| 20–53 μm | 66.3 | 31.7 | 2.0 | 0.26 |
| 45–90 μm Ø | 46.7 | 52.2 | 1.1 | 0.25 |
| 45–90 μm | 64.6 | 33.3 | 2.1 | 0.20 |

Juan [78] performed experiments on SS 316L with three different powder size of SS 316L with a mean particle size of 15, 24, and 41 μm. Printing was performed on M-Flex ExOne. The printing parameter included recoated speed of 130 mm/s, roller speed of 250 rpm, roller traverse speed of 15 mm/s and drying speed of 17 mm/s. Sample dimensions were $0.5 \times 0.5 \times 0.2$ in$^3$. After printing, samples were cured at 195 °C in a furnace for 8 h. Before sintering, samples were debinded at 600 °C in a vacuum furnace. Sintering was conducted in 96% Ar–4% $H_2$ environment to prevent oxidation.

Samples were sintered in 1300 °C, 1356 °C, and 1380 °C and held for 90 min, 360 min, and 1440 min at each temperature, respectively.

Results of Juan [78] showed that smaller particle size resulted in higher final density in most cases. The highest density of 92.35% was related to the smallest particle size (15 μm) sintered at 1356 °C. According to Herring's scaling law [101] under the same experimental condition, the required time for the larger powder to get the same degree of sintering is longer than that for smaller particles. Furthermore, for the same sintering temperature GB diffusion governs the sintering mechanism for finer powders and is faster compared to volume diffusion as dominant sintering mechanism in larger particles [93]. Generally, smaller particles have smaller gaps and pores, which can be eliminated more easily compared to larger pores.

*8.2. Effect of Sintering Time and Temperature*

Verlee et al. [56] conducted experiments on the effects of sintering temperature and time on the final part properties. Sintering was performed in a batch furnace in a hydrogen environment and held for 90 min at different temperatures (varied from 1200 °C–1430 °C) or at a fixed temperature for different holding times. Density, open and closed porosity were determined by the Archimedes method.

$$\text{Open porosity}: \; op = \frac{m_3 - m_1}{m_3 - m_2} \tag{7}$$

$$\text{Closed porosity}: \; cp = 1 - \frac{\rho_{fluid}}{m_3 - m_2}\left(\frac{m_1}{\rho_{material}} + \frac{m_3 - m_1}{\rho_{fluid}}\right) \tag{8}$$

where $m_1$ is the weight of sample in the air, $m_2$ sample mass minus the Archimedes buoyancy, $m_3$ sample mass after taking from solvent (corresponds to $m_1$ with the addition of the mass of solvent contained in the open pores).

Verlee et al. [56] reported that final density depends on particle size and sintering temperature. For a selected powder, higher density was achieved at higher sintering temperature. Furthermore, final density depends on the green density too. Higher final density (98.7%) was obtained from a relatively larger sample (31 μm) with higher green density, whereas the smallest sample (22 μm) demonstrated only 97.5% of final density. Non-spherical powders obtained the lowest final density as started from a lower green density.

Densification highly depends on sintering temperature because at temperature >1200 °C austenite (FCC) transforms to delta-ferrite (BCC), which has a greater diffusion rate. Thus, densification is improved. At temperature >1395 °C, densification will be even further improved due to the presence of a liquid phase. The capillary forces from the liquid phase act on the solid state to eliminate the porosity and consequently reduce the interfacial area [92].

Table 7 summarizes final density, range of open and closed porosity, and densification of samples sintered at 1395 °C for 90 min. For all samples, open porosity decreased as density increased. Densification ($\psi$) decreased as mean particle size increased, regardless of powder shape. However, particle shape had a high effect on final density and range of porosity. Irregular particles resulted in lower final density and higher porosity. Generally, the higher the sintering temperature, the higher the final density, and the lower the connected or open porosity were achieved [56].

Sintering time was varied from 1–300 min at sintering temperatures of 1315, 1335, and 1345 °C. Results of Verlee et al. [56] showed that final density was a linear function of sintering time up to 89% at a given sintering temperature, then densification rate decreased. The final stage of sintering was not obtained until holding for 90 min at each sintering temperature. Sintering profile, including time and temperature, dictates the pore range (open and closed porosity percentage) within the final part [56].

Johnston et al. [108] investigated the strain or shrinkage at the initial stage of sintering for binder jetted SS 316L powder with an average size of 80 µm printed with a polymetric-based binder on a ProMetal RTS300 printer. Dimensions of $5 \times 5 \times 20$ mm$^3$ were considered for the samples with 20 mm on the fast print axis. The initial strain was zero due to no compaction force on powders. Samples then were cured at 205 °C for four hours. Sintering was performed at 96% Ar–4% H$_2$ environment with 28 L/h flow rate in a Netzsch Dilatometer 402C using a TASC 414/3 controller. Samples were heated at 5 °C/min and held for 30 min at 200 °C and 465 °C to completely cure and debind the binder, respectively. After holding at 465 °C, samples were heated with a heating rate of 4 °C/min, 7 °C/min, 10 °C/min, and 20 °C/min to peak temperatures of 1010, 1100, 1180, and 1263 °C, respectively and held for 2 h at the peak temperatures. Shrinkage was measured using dilatometer, and the results revealed that heating rate significantly affects the strain formation during ramp up and subsequent isothermal intervals. Strain formation increased dramatically during ramp-up intervals at temperature >1120 °C. Higher heating rate led to greater strain formation during subsequent isothermal intervals. Furthermore, the amount of strain produced at peak temperature increased with peak isothermal temperature.

Wang et al. [14] investigated the effects of sintering parameters on the linear shrinkage in *x*, *y*, and *z*-axis of BJ manufactured 316L SS parts. Taguchi method was exploited to reduce the cost of the experiment. Default sintering profile was adopted from Exone. Isothermal sintering temperature, heating rate and sintering time were considered as design factors in this study. For each factor, three levels were defined, as shown in Table 8 [14].

**Table 8.** Experimental factors with each level [14].

| Factors | Level 1 | Level 2 | Level 3 |
|---|---|---|---|
| Factor A: Sintering temperature (°C) | 1010 | 1100 | 1190 |
| Factor B: Sintering rate (°C/min) | 4 | 12 | 20 |
| Factor C: Sintering time (min) | 60 | 90 | 120 |

The analysis determined optimal sintering parameters for x, y, and z-axis as well as percentage improvement of dimensional accuracy after applying optimal parameters for all three axes. Additionally, considering all three axes simultaneously, one set of optimal sintering parameters has been selected. Table 9 summarizes the optimal sintering parameters along with percentage improvement of dimensional accuracy for all three axes.

**Table 9.** Optimal parameters of sintering profile and percentage improvement of dimensional accuracy [14].

| Optimal Parameters | Sintering Temperature (°C) | Heating Rate (°C/min) | Sintering Time (min) | Percentage Improvement |
|---|---|---|---|---|
| For dimensional accuracy of *x*-axis | 1010 | 12 | 60 | 54.66% |
| For dimensional accuracy of *y*-axis | 1010 | 12 | 90 | 32.86% |
| For dimensional accuracy of *z*-axis | 1190 | 12 | 60 | 73.98% |
| For considering dimensional accuracy of all three axes | 1100 | 12 | 60 | for *x*-axis: 45.34% for *y*-axis: 3.29% for *z*-axis: 32.29% |

Analysis of variance (ANOVA) is a performance measurement method which statistically indicates at what percentage the factor influences the performance characteristic. ANOVA was also performed to indicate the most critical factor on the dimensional accuracy of all three axes. Based on ANOVA result, sintering temperature had the highest percentage contribution on the shrinkage rate among all design factors. Thus, sintering temperature was the most critical factor on the dimensional accuracy of all three axes [14].

Juan [78] investigated the effect of sintering temperature and time on densification of three different particle size of 15, 24, and 41 μm of SS 316L powder. According to sintering theory, densification will increase with sintering temperature [63,93,96]. However, results [78] showed a reduction in the final density of parts at temperature >1356 °C. The microstructure of SS 316L parts after sintering at 1300–1380 °C for 90 min revealed pores that were attached to grain boundaries (GB) at sintering temperature of 1300 °C with a slow rate of grain growth. In this type of microstructure, pores would move together with grain boundaries as grains grow until eventually the pores impede grain growth. With increasing temperature to 1356 °C, pores shrank and became isolated within the grains. So, the total free energy of the system increased as the pores and GBs became separated, and new interfacial areas were created [93].

Dominant diffusion mechanism was found to be volume diffusion since the relative grain growth rate was higher than the relative densification rate [93,112]. Because the rate of volume diffusion was much slower than that of grain boundary diffusion, the isolated pores within grains prohibited the full densification. By raising the temperature, pores became larger and emerged on the grain boundaries. The total number of pores did not decrease, whereas the size of pores increased at elevated temperatures. Therefore, increasing sintering temperature (T > 1356 °C in Juan's study [78]) did not improve the densification any further. Because, the pressure of entrapped gas within isolated pores became equal to the capillary pressure of the pore in a way that pore stopped shrinking, and maximum achievable sintered density was reached [93,113,114]. However, sintering in a vacuum could eliminate the pores during the final stage of sintering [115].

Juan et al. [78] investigated the impact of sintering holding time on the final density of samples. Samples were held for 90 min, 360 min, and 1440 min at 1300, 1356, and 1380 °C. Results revealed that for samples sintered at 1300 °C, increasing holding time improved the final density. While samples sintered at 1356 °C attained their highest density with 360 min holding time. Increasing time after this point did not improve the sintered density any further. This can be attributed to pore coarsening without a reduction in the number of pores when increasing the holding time at 1356 °C.

### 8.3. Effect of (Sintering) Additives

Rego et al. [55] investigated the effect of different additives at different sintering temperatures on the final density and mechanical properties of the parts. The mixture of the sample (70L-25M-5S) (shown in Table 4) with the highest green density of 63.87% were mixed with 0.25, 0.5, 0.75 wt.% of boron (B), boron carbide (BC), and boron nitride (BN), respectively as shown in Table 10. Then, all samples with and without additives were printed and sintered in a vacuum environment at sintering temperatures of 1200 °C, 1300 °C, and 1350 °C for 6 h. After sintering, samples with different amounts of additives obtained different sintered densities. Figure 17 presents the final density of samples printed with different amounts of additives followed by sintering at different temperatures.

**Table 10.** List of sintering additives, their particle size, and wt.%. Boron (B), boron carbide (BC), and boron nitride (BN).

| Powders | | Mean Particle Size (μm) [55] | Wt.% |
|---|---|---|---|
| Sintering Additives | B | 1 | 0, 0.25, 0.5, 0.75 |
| | BN | 1 | 0, 0.25, 0.5, 0.75 |
| | BC | 0.6 | 0, 0.25, 0.5, 0.75 |

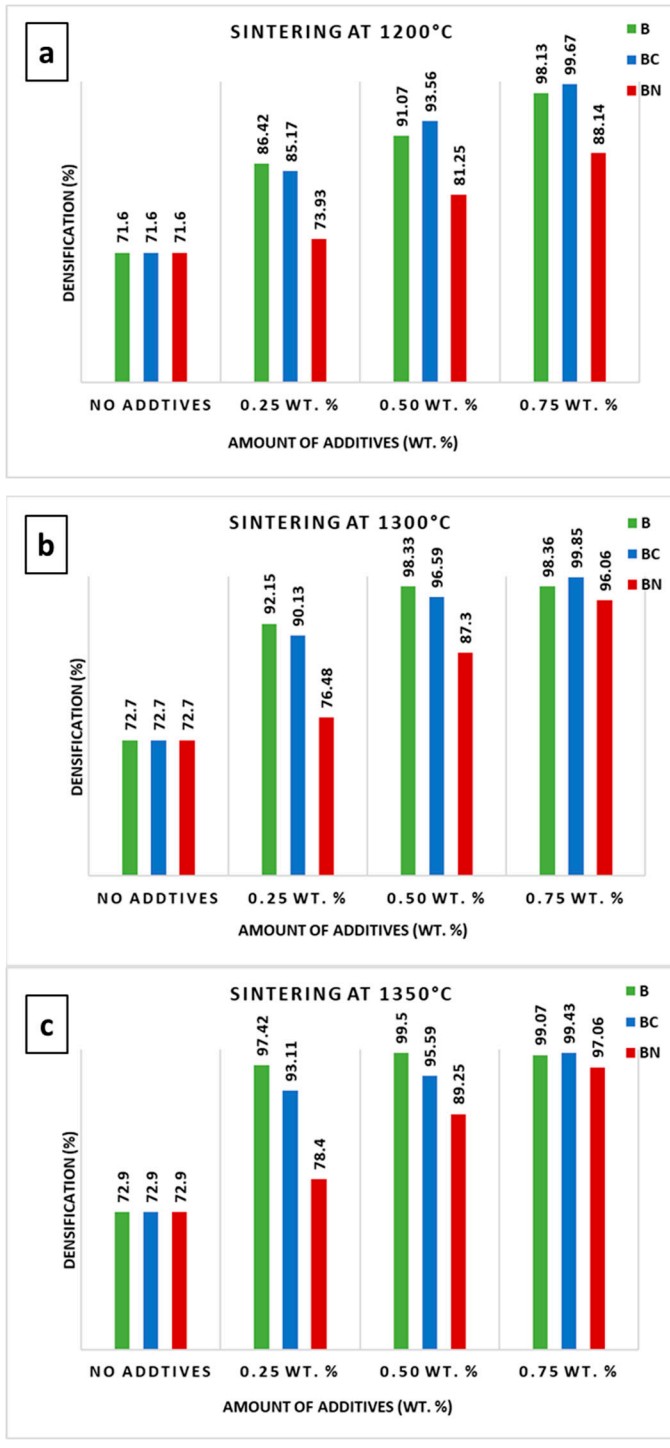

**Figure 17.** Densities of samples with different amounts of additives sintered at (**a**) 1200 °C, (**b**) 1300 °C, and (**c**) 1350 °C.

Although some samples obtained very high sintered density (>99.85%), the distortion was not in the acceptable range. Samples with the highest density and least shrinkage rate are listed in Table 11. The microstructure of samples with different amounts of additives demonstrated different porosity. Samples sintered at 1300 °C without additive and with 0.25%B demonstrated large pores, while samples sintered at the same temperature with 0.5%B and 0.75%B exhibited dense structures with no apparent large-size pores. Samples sintered at 1200 °C with 0.75%B and 0.75%BC showed relatively high density, but few pores with noticeable size were observed.

**Table 11.** 316L SS sintered samples with the highest final relative density.

| Additives (wt.%) | Sintering Temperature (°C) | Final Relative Density (%) [55] |
|---|---|---|
| 0.75% BC | 1200 | 99.67 |
| 0.5% B | 1300 | 98.33 |
| 0.75% B | 1200 | 98.13 |

### 8.4. Effect of Hot Isostatic Pressing

In addition to standard heating process after printing, hot isostatic pressing (HIP) can improve the final density of the part. Frykholm et al. [27] printed SS 316L using standard Digital Metal processing and sintered at 1380 °C in a vacuum with the partial pressure of Ar to prevent Cr loss from the surface. Sintered samples achieved 96% relative density. To obtain higher density, HIP was performed at 1150 °C for 1.5 h with a pressure of 100 MPa. Figure 18 shows the microstructure of sintered and HIPPed SS 316L samples. After the HIP process 316L obtained near full density.

As a requirement before the HIP process, sintered density should have been high enough with no open pores since only porosities that are fully subsurface will collapse. Otherwise, HIP will not be effective in improving the density in case of open (surface connected) porosity because the pressure is uniformly applied through the part on all of the pore surfaces. However, for closed pores, the pressure will be applied uniformly only on the outer specimen surfaces leading to full densification.

Table 12 shows how the mechanical performance of the printed and sintered SS 316L was comparable to those for samples produced via metal injection molding (MIM).

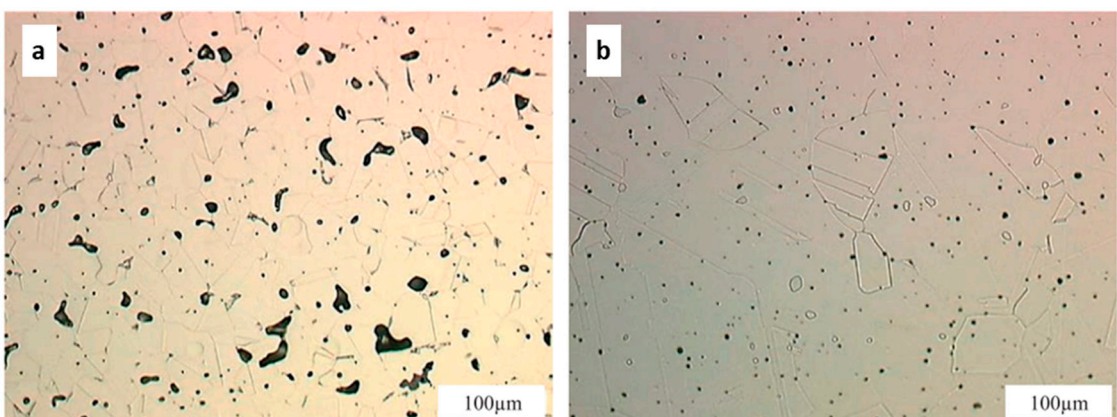

**Figure 18.** Optical micrographs showing pores structure for (**a**) sintered and (**b**) sintered and HIPed SS 316L [27].

**Table 12.** Mechanical performance of 316L for BJ printed and metal injection molding (MIM) samples [27].

| Samples | Density (g/cm$^3$) | Tensile Strength (MPa) | Yield Strength (MPa) | Elongation (%) | Hardness (HRB) |
|---|---|---|---|---|---|
| BJ 316L | 7.67 | 511 | 170 | 58 | 60 |
| MIM 316L | 7.6 | 520 | 175 | 50 | 67 |

### 8.5. Effect of Sintering Atmosphere

Juan [78] reported on the effect of sintering atmosphere on the final density of parts. Samples were sintered in the vacuum and 96% Ar–4% H$_2$ atmosphere at 1300 °C for 90 min. As shown in Figure 19, samples sintered in vacuum achieved 10% higher density compared to those sintered at Ar environment for the same sintering temperature. In terms of particle size effect, vacuum sintering showed the same

results as Ar sintering in which finer particles obtained higher density. Comparing the microstructure of samples sintered at vacuum and Ar at 1300 °C for three different particle sizes of 15, 24, and 41 μm revealed that samples sintered at Ar involved the channel of pores along grain boundaries, that could retard the grain growth and create irregular-shaped grains. Presence of entrapped gas in the pores decreased the pore shrinkage rate and prevented full densification.

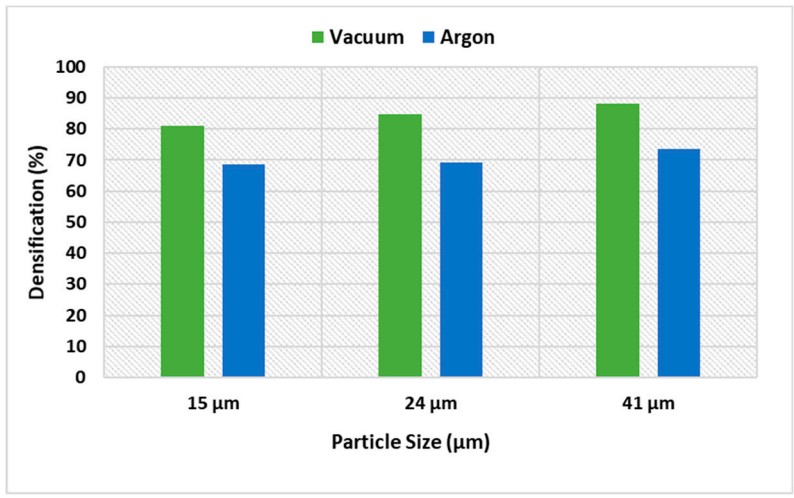

**Figure 19.** Comparison of vacuum and argon atmosphere influencing the densification with different particle size.

*8.6. Mechanical Properties*

Mechanical properties of BJ manufactured parts have not been extensively investigated. Nastac et al. [116] fabricated 316L SS parts using Exone printer. The microstructure of the printed sample revealed that BJ process created relatively fine equiaxed grain microstructure as opposed to columnar grain microstructures generated in laser-based AM processes like SLM [116]. Advantage of equiaxed grain structure is generating isotropic properties, and thus, the parts demonstrate similar mechanical properties in X, Y, and Z directions, regardless of the built direction [117,118]. Lack of chemical segregation at the grain boundaries is another advantage of BJ, because no melting is involved in BJ process. Therefore, microstructure analysis and evaluated mechanical properties suggested that mechanical properties of the built samples were affected by the size, shape, and distribution of the pores, as well as by the size and shape of the grains [116].

Mechanical properties of the produced 316L sample using BJ [116] are compared with the ASTM-B883 Standard Specification for MIM Materials and presented in Table 13. Thus, manufactured 316L part demonstrated equivalent values for tensile and hardness compared to the MIM sample. BJ manufactured parts had a higher yield strength than standard MIM part with less elongation.

**Table 13.** Mechanical properties of 316L for manufactured BJ part and MIM materials (American Society for Testing and Materials (ASTM)-B883).

| Samples | Tensile Strength (MPa) | Yield Strength (MPa) | Elongation at Break (%) | Hardness (HRB) | Density (g/cm$^3$) |
|---|---|---|---|---|---|
| BJ 316L [116] | 517 | 214 | 43 | 66 | 7.7 |
| MIM 316L (as-sintered) [119] | 520 | 175 | 50 | 67 | 7.6 |

Rego et al. [55] measured hardness of the SS 316L samples after adding different amount of additives and sintered at different temperature. Details related to additives and sintering temperature are formerly provided in Table 10 and Figure 17. The average hardness of samples is shown in Figure 20a–c. Overall, samples with B additive demonstrated higher hardness compared to samples with BN and BC additives. Samples with the highest amount of boron, 0.75%B, also had the highest hardness of 94.17 HRB after sintering at 1350 °C and 1300 °C which is higher than 80 HRB being reported to be the hardness for the commercially obtained SS 316L bar [55]. Inconsistency in hardness values may be attributed to a different level of porosity in the samples.

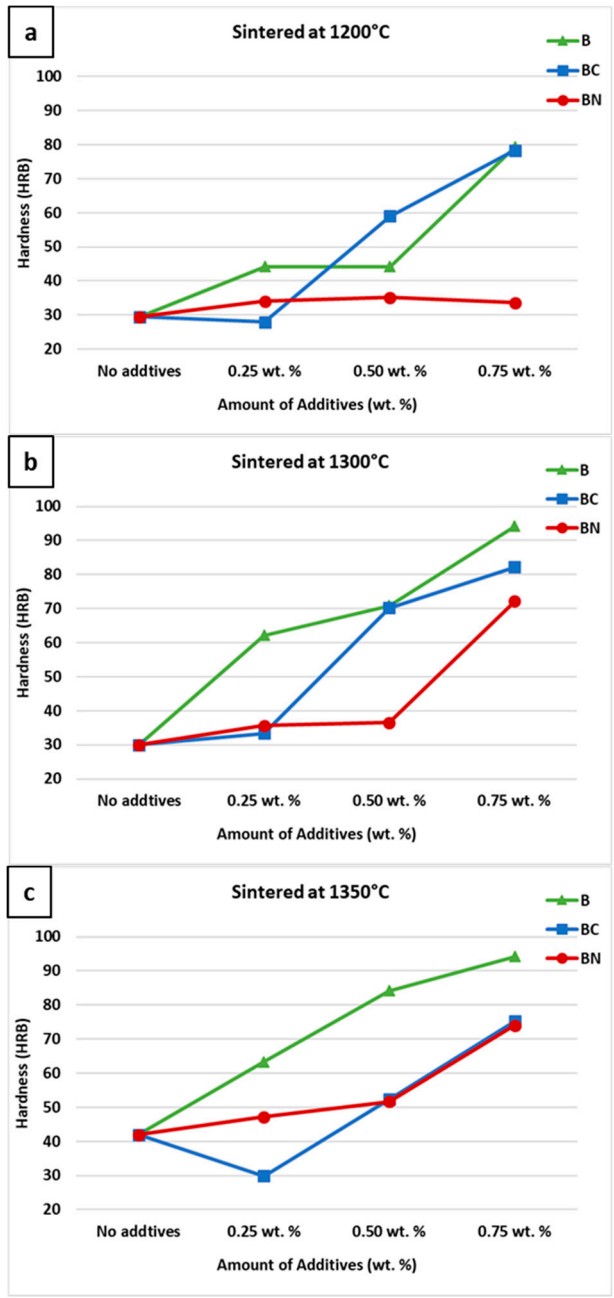

**Figure 20.** Hardness of samples with different amount of additives sintered at (**a**) 1200 °C, (**b**) 1300 °C, and (**c**) 1350 °C.

Verlee et al. [56] evaluated tensile strength and elongation on tensile bars formed from 31 µm and 20–53 µm powders and sintered at different temperatures for 90 min. Table 14 summarizes maximum strength and elongation of samples fabricated from two different powders with various final density and sintered at different temperatures. Results revealed that mechanical properties were affected by final density, particle size, and pore size. Maximum strength and elongation increased with final density. Also, samples sintered at higher temperature demonstrated higher strength and elongation that is attributed to a lesser percent of porosity of samples. Samples sintered at higher temperature had higher final density and maximum strength. Porosity reduction can highly improve the mechanical properties of parts since pores are high-stress concentration defects that can initiate or promote crack growth [4].

**Table 14.** Maximum strength and elongation of sintered tensile bars built with 31 µm or 20–53 µm powder [56].

| Powders | Sintering Temperature (°C) | Final Density (%) ±0.1% | Max. Strength (MPa) ±6% | Elongation (%) ±5% |
|---|---|---|---|---|
| 31 µm | 1255 | 81.0 | 309 | 21.3 |
| | 1335 | 85.2 | 388 | 35.5 |
| | 1365 | 90.9 | 437 | 52.1 |
| | 1395 | 98.0 | 518 | 61.9 |
| 20–53 µm | 1415 | 81.8 | 243 | 25.4 |
| | 1432 | 88.2 | 310 | 29.9 |

Zhou et al. [90] introduced Instrumented Indentation Testing (IIT) to measure Young's modulus from the free surface of parts fabricated by BJ, especially for bone implants. The IIT system utilizes a specially designed probe tip to indent the surface of the sample, and at the same time measuring the applied force and the displacement. Young's modulus was evaluated for a solid cylinder, cylinder of 1.0 and 1.5 mm lattice structure (Figure 21) using IIT, 3-point bending test, and compression test. Reported data which are summarized in Table 15, were a value of 4.07, 0.446, and 1.50 GPa for solid, grid lattice sizes of 1 and 1.5 mm, respectively. Measured values were significantly less than 200 GPa [90,120] for a conventionally manufactured 316L SS. The IIT bears a 15% error measuring solid and lattice freeform samples, compared to the benchmark results gained from traditional testing machines and the 3-point bending.

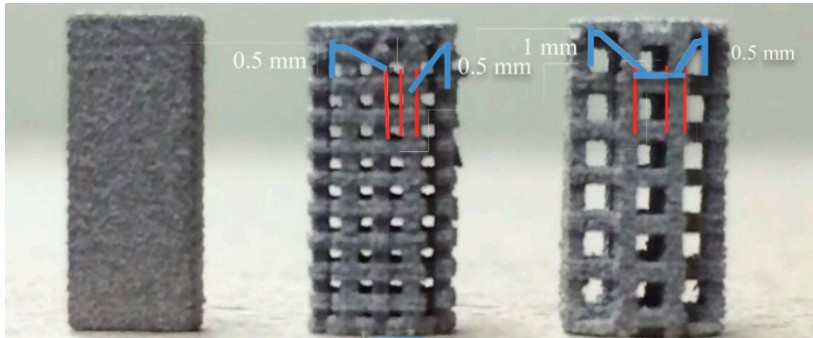

**Figure 21.** Solid, grid lattice of 1.0 mm, grid lattice of 1.5 mm structure [90].

**Table 15.** Young's modulus and relative density for three different samples manufactured by BJ.

| Samples | Standard | Young's Modulus (GPa) [90] | | | Theoretical Density (g/cm³) | Measured Relative Density (%) |
| | | Instrumented Indentation Testing (IIT) | 3-Point Bending Test | Compression Test | | |
|---|---|---|---|---|---|---|
| Conventional 316L SS | 200 [120] | | | | 8 | 100 |
| BJ Printed Parts | | | | | | |
| Solid | | 4.07 | 4.4 | 4.07 | 4.19 | 50 |
| 1.0 mm Lattice structure | | 1.5 | 1.41 | 1.5 | 2.1 | 25 |
| 1.5 mm Lattice structure | | 0.446 | 0.46 | 0.446 | 1.1 | 12.5 |

Shrestha et al. [107] employed Taguchi method to determine optimum printing parameters to improve TRS of manufactured 316L samples using BJ. Binder saturation, layer thickness, roller speed, and feed-to-powder ratio (thickness of feed layer to layer thickness) were considered as design factors in this study. Binder saturation and layer thickness affect binder-powder interaction [121] and, roller speed and feed-to-powder ratio influence packing density of each layer [122]. For each factor, three levels were defined as shown in Table 16.

**Table 16.** Printing parameters at three levels [107].

| Factors | Level 1 | Level 2 | Level 3 |
|---|---|---|---|
| Factor A: Binder saturation (%) | 35 | 70 | 100 |
| Factor B: Layer thickness (µm) | 80 | 100 | 120 |
| Factor C: Roller speed (mm/s) | 6 | 10 | 14 |
| Factor D: Feed-to-powder ratio | 1 | 2 | 3 |

Standard ASTM B528-99 specimens of 316L were printed on Exone X-1 lab. Samples were cured at 190 °C for 4 h. Then samples were sintered in vacuum at 1120 °C for 2 h. Instron 5500R, Norwood, MA, USA was used for testing. The values shown in Table 17 were found to be the optimal level for each printing parameters. ANOVA analysis was performed to determine the significance of the parameters. It was identified that binder saturation and feed-to-powder ratio had a remarkable influence on the mean TRS values [107]. The predicted long-range mean for TRS was calculated to be 83.34 MPa. The estimated optimum TRS was 83 ± 7 MPa. Considering the optimal parameters, the TRS value increased from 65.32 to 90.59 MPa by 38.69% [107].

**Table 17.** Optimization of binder jetting parameters using Taguchi method.

| Factors | | Optimal Level [107] |
|---|---|---|
| Factor A | Binder saturation (%) | 70 |
| Factor B | Layer thickness (µm) | 100 |
| Factor C | Roller speed (mm/s) | 6 |
| Factor D | Feed-to-powder ratio | 3 |

The surface roughness of BJ printed parts depends on printing resolution, layer thickness, and powder size [15,28,36,55]. It has been reported that printing mono-size powder led to higher surface roughness, while multi-modal powder size resulted in lower surface roughness and better packing ratio. It was attributed to filling gaps by smaller particles and creating a smoother surface. Also, the addition of additives like boron to multi-modal powder can significantly improve the surface finish of BJ manufactured parts [55].

## 9. Summary

Binder jetting technology has been developed to manufacture functional products rather than only prototyping purposes. It has a wide range of applications in various fields and industries including in the biomedical, chemical, and automotive industries. Binder jetting of SS 316L was reviewed in this article for the first time. Because it is of one of the most applicable stainless steel in industrial applications and most studied alloy in currently published literature. Reported literature revealed that mono-size 316L gas atomized powder with spherical shape and average particle size of 22–30 μm is an optimal feedstock to achieve the full density. Utilizing multi-modal powder size along with sintering additives such as boron up to 0.75 wt.% could significantly improve the final density up to 99.67% relative density.

Printing parameters influence the part quality and surface finish. Most acceptable parts were manufactured by layer thickness around 50–80 μm and binder saturation of 60–70%. Sintering profile including temperature, time, and atmosphere proved to be highly effective on densification and mechanical behavior of components. Many factors determine the optimal sintering profile. However, increasing temperature to 1360–1380 °C for the peak isothermal and holding time of 4–6 h in a vacuum or reducing atmosphere will noticeably enhance part properties.

By utilizing post-processing, including HIP process full density is attainable. Although valuable experiments have been conducted to report the impact of feedstock and sintering profile on final part properties of SS 316L, a limited number of articles have dedicated their experiments to evaluate the mechanical behavior of BJ components. Therefore, detailed measurement and analysis of mechanical properties of BJ manufactured SS 316 is currently lacking in the literature. BJ manufactured parts exhibited higher or equivalent hardness compared to conventionally manufactured parts. Maximum strength and elongation at break increased with final density.

The contribution of this article is to review and summarize BJ manufactured SS 316L parts and to provide a comparison with conventionally manufactured SS 316L parts. The aim of this review article is to provide comprehensive guidance to select optimal process parameters for all BJ steps including powder feedstock properties, printing parameters, and post-processing set-up. These optimal conditions will lead to achieving full density and desired properties in BJ manufactured parts.

## 10. Future Direction in Binder Jetting of 316L

While numerous steels are fabricated by conventional manufacturing, including forging, casting, and machining, there is a limited number of steel parts that have been developed by additive manufacturing [41]. Thus, more efforts are required to develop and adopt new alloys in AM. SS 316L is the most popular steel used in BJ by both industries and researchers [27]. Current research are focused on investigating the impact of feedstock (e.g., powders) properties [55,56], printing parameters including layer thickness and binder saturation [107], and post-processing parameters including sintering time and temperature on final properties of 316L parts [56,78]. Despite numerous studies in the topics mentioned above, many aspects of BJ processes are still poorly understood. For instance, only a few successful fully dense 316L parts [27] have been reported in the literature. Therefore, BJ post-processing still needs more improvements to obtain a full density part. Furthermore, lack of a library for selecting suitable binders for specific metal powder and the effect of binder characteristics on final part properties needs to be addressed in the future research. There is a very limited number of studies on the mechanical properties of BJ 316L. Future studies should be focused on reporting repeatable and reproducible mechanical properties of BJ 316L including hardness and yield strength.

The future studies would need to generate a reliable set of data on comparison of mechanical properties of developed BJ 316L parts from different feedstocks (pure 316L as opposed to 316L + additives) with different printing parameters (binder saturation and drying time). Consequently, current gaps will be filled, and the performance of BJ 316L parts in various industrial the application will be evaluated.

**Funding:** The funding of equipment provided by the Murdock Charitable Trust under contract #2016231: MNL: 5/18/2017.

**Acknowledgments:** The authors would like to acknowledge the Oregon Business Development Department for the High Impact Opportunity Projects (HIOP) for financial support. Authors would also like to thank Hewlett-Packard Company in Corvallis and Advanced Technology and Manufacturing Institute (ATAMI) director and staff for their support.

**Conflicts of Interest:** The authors declare no conflicts of interest.

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
