# Peer review of "A Review on Binder Jet Additive Manufacturing of 316L Stainless Steel"

_jmmp, doi:10.3390/jmmp3030082_

Round 1
Reviewer 1 Report
Abstract:
· The English needs to be improved
· It is unclear to find the novelty in the manuscript. What has been the main outcome of the extensive literature review?
· It is difficult to pin point the novelty in this paper
Introduction:
· More references need to be added specially the ones related to the construction sector: https://doi.org/10.1016/j.autcon.2018.05.005
· Throughout the sections of the paper more interesting and novel subsections need to be presented.
· The gaps in the knowledge has not been emphasised on as it is expected from a review paper.
The sections in the manuscript do not follow a logical structure. The structure of the manuscript requires revision
Section 3: materials is very vague and not detailed
All the tables and figures are from previous literature. There is only one (Table 6) author original culmination of previous efforts.
Some of the figures are really low quality and not appropriate for publication (e.g. Figure 22, 27)
Section 6.4: why is the Hot with capital H? or section 6.6.5 the word Roughness with capital R
Author Response
7/16/19
To the Editor of Journal of Manufacturing and Materials Processing:
We are very excited to have been given the opportunity to revise our review article titled “A Review on Binder Jet Additive Manufacturing of 316L Stainless Steel.” We would like to thank the reviewers for the careful and thorough reading of this manuscript. We express special gratitude to reviewers for the thoughtful comments and constructive suggestions, which helped us tremendously to improve the quality of this manuscript. We carefully considered all of the reviewers’ comments and addressed them in the best possible way. Herein, we explain how we revised the paper based on those comments.
In particular, section 2 is added to the revised manuscript on current status of binder jetting, novelty and contribution of this review article and the existing gaps in the literature. More tables and figures are generated by authors as original plots. The permission for other published data (figures and tables) are acquired from the corresponding publishers. The abstract and summary are revised to address the reviewer’s comments. Section 5 is added to the revised manuscript. Section 6 is revised accordingly. For the ease of reading, the comments are typed in black and our response are typed in blue. Please let us know if you need any further information.
While there are very few review papers published on binder jetting, there is NOT any review paper on metal binder jetting in the current literature. This manuscript, if accepted, will be the FIRST review article on metal binder jetting and will bring a significant number of citations and visibility to the esteemed Journal of Manufacturing and Materials Processing.
Permanent coordinates for the corresponding author are as follows: Oregon State University, School of Mechanical, Industrial, and Manufacturing Engineering, 2000 SW Monroe Ave, 204 Rogers Hall, Corvallis, OR 97331, 541-737-3685 and [email protected]
Thank you in advance for considering our revised manuscript.
Respectfully,
Somayeh Pasebani, PhD, PMT
Assistant Professor of Advanced Manufacturing
School of Mechanical, Industrial, and Manufacturing Engineering, Oregon State University
Corvallis, OR 97331-6001
Reviewer 1
Comments and Suggestions for Authors
Abstract:
1. The English needs to be improved
Our Response:
The entire text of the article is reviewed to address this comment. In particular the abstract is accordingly revised with better English.
2. It is unclear to find the novelty in the manuscript. What has been the main outcome of the extensive literature review?
Our Response:
We appreciate the comments of the reviewer 1. We agree with the reviewer’s comments and therefore, one section is added to the article to explicitly express the current BJ status, the contribution and novelty of this work and the existing gap in the literature (Section2) as follows:
2. Current Status of Binder Jetting Technology
Metal binder jetting is promising to lower manufacturing cost and lead time for complex geometry and design compared to the conventional manufacturing method. One of the most common alloys in numerous industries is SS 316L which has been studied by many researchers. However, breakthroughs in BJ of SS 316L has not been summarized and reviewed yet to the best knowledge of authors. The aim of this article is to provide guidance for conducting experiments and improving parameters/set-up to build functional 316L parts with competitive properties to wrought or pressed and sintered SS 316L parts. To achieve this, the current status of published studies is reviewed and summarized.
The current status of binder jetting of SS 316L (as will be shown later in Table 3) provides an extensive data on selection of feedstock (e.g., powders) and sintering profiles to obtain near full density that are reviewed in this article. However, there is a gap in the current literature for selecting the optimal binder and identifying the role of binder on the final density and dimensional accuracy of the parts. Furthermore, mechanical properties in current literature need to be evaluated and analyzed intensively to report reliable and repeatable data. The authors of this review article highlight the existing gaps in the literature and provide a comprehensive summary and comparison in the field of BJ, in particular, SS 316L.
Furthermore, novelty, contribution of this review article has been added to summary section too:
The contribution of this article is to review and summarize BJ manufactured SS 316L parts and to provide a comparison with conventionally manufactured SS 316L parts. The aim of this review article is to provide comprehensive guidance to select optimal process parameters for all BJ steps including powder feedstock properties, printing parameters, and post-processing set-up. These optimal conditions will lead to achieving full density and desired properties in BJ manufactured parts.
Furthermore, current gaps in the literature are highlighted in this review article. For instance, there is a gap in the current literature for selecting the optimal binder and identifying the role of binder on the final density and dimensional accuracy of the parts. As another example, a detailed investigation of mechanical properties (at low and elevated temperatures) and microstructural characterization in BJ manufactured SS 316L deem necessary. In the same way, the prediction of microstructure based on processing and post-processing parameters are currently lacking.
To the best knowledge of authors, there has not been any review paper published on BJ of metallic parts. Thus, it is the novelty of this article to provide such a comprehensive review of BJ of SS 316L that correlates powder feedstock properties, printing parameters, and post-processing parameters to part density, microstructure, and mechanical properties.
3. It is difficult to pin point the novelty in this paper
Our Response:
Novelty of this work has been added to both section 2 and summary:
The contribution of this article is to review and summarize BJ manufactured SS 316L parts and to provide a comparison with conventionally manufactured SS 316L parts. The aim of this review article is to provide comprehensive guidance to select optimal process parameters for all BJ steps including powder feedstock properties, printing parameters, and post-processing set-up. These optimal conditions will lead to achieving full density and desired properties in BJ manufactured parts.
Furthermore, current gaps in the literature are highlighted in this review article. For instance, there is a gap in the current literature for selecting the optimal binder and identifying the role of binder on the final density and dimensional accuracy of the parts. As another example, a detailed investigation of mechanical properties (at low and elevated temperatures) and microstructural characterization in BJ manufactured SS 316L deem necessary. In the same way, the prediction of microstructure based on processing and post-processing parameters are currently lacking.
To the best knowledge of authors, there has not been any review paper published on BJ of metallic parts. Thus, it is the novelty of this article to provide such a comprehensive review of BJ of SS 316L that correlates powder feedstock properties, printing parameters, and post-processing parameters to part density, microstructure, and mechanical properties.
Introduction:
4. More references need to be added specially the ones related to the construction sector: https://doi.org/10.1016/j.autcon.2018.05.005
Our Response:
Thank you for offering this article. It is cited in the related section (Introduction) as Ref. [13].
5. Throughout the sections of the paper more interesting and novel subsections need to be presented.
Our Response:
This comment was very helpful and constructive. We appreciate it a lot. Therefore, sections and subsections are revised. Throughout the paper many section and subsection are re-entitled and some subsections are added. For example, section 2, section 5.1, section 6.4.3 and section 6.5 are added to the revised manuscript to address this comment.
6. The gaps in the knowledge has not been emphasized on as it is expected from a review paper.
Our Response:
Current gap in the literature is addressed in both section 2 and summary:
Furthermore, current gaps in the literature are highlighted in this review article. For instance, there is a gap in the current literature for selecting the optimal binder and identifying the role of binder on the final density and dimensional accuracy of the parts. As another example, a detailed investigation of mechanical properties (at low and elevated temperatures) and microstructural characterization in BJ manufactured SS 316L deem necessary. In the same way, the prediction of microstructure based on processing and post-processing parameters are currently lacking.
7. The sections in the manuscript do not follow a logical structure. The structure of the manuscript requires revision
Our Response:
Entire the article is reviewed and revised accordingly with better structure and headings.
8. Section 3: materials is very vague and not detailed paper.
Our Response:
Materials section (currently section 4.) is thoroughly revised and explained in more details. Also, subsections follow a clearer structure. For example, the subheadings in section 4 are currently as follows:
4.1. Stainless Steel 316L Composition and Properties
4.2. Common Defects of SS 316L Components at Elevated Temperatures
4.3. Additively Manufactured SS 316L Parts
9. All the tables and figures are from previous literature. There is only one (Table 6) author original culmination of previous efforts.
Our Response:
Many of the presented tables and figures are generated by authors and entire the table/figure is cited. Thus, edition and revision of citations is applied accordingly. Tables 1, 4, 10, 11, 13, 15, and 17 are generated by current authors as original work. Also, several of figures/tables are replaced by the figures generated by current authors such as Figures 11, 17, and 19.
10. Some of the figures are really low quality and not appropriate for publication (e.g. Figure 22, 27)
Our Response:
Figure 22 is removed accordingly, and figure 27 (currently figure 19) is replaced with the one generated by author with a better quality.
11. Section 6.4: why is the Hot with capital H? or section 6.6.5 the word Roughness with capital R
Our Response:
The entire article is revised for titles and subsections. All the section and subsections are written with the capital first letter (applied to all words in the title lines). That is why “R” is capital. This is now fixed.
Reviewer 2 Report
Review on
“A Review on Binder Jet Additive Manufacturing of 316L Stainless Steel”
by
Mirzababaei and Pasebani
(manuscript jmmp- 508435)
Submitted June 2019 to “Journal of Manufacturing and Materials Processing”
In this paper the authors give an extensive review on binder jetting of 316L stainless steel. Up to date and relevant literature on materials and processing is considered in this well-written paper. The addressed topic is of great interest and in scope of the journal. I only have few remarks, which should be addressed by the authors in a revised version of the article. After incorporation of the proposed changes, the paper should be ready for publication. Below please find my remarks and suggestions for improvement.
Comments:
The discussion in section 4.1. “Powder Size and Shape” on bulk solids characteristics and spreadability, respectively, sinterability should be re-written and expanded. In the section, there are several flaws, for example, one cannot generalize that smaller particles have less surface roughness than larger one. Moreover, the discussion on the usage of bimodal distributed particles and wet vs. dry powder application seems somehow sloppy. In both cases, I would argue via the particle size dependency and dependency of the surrounding medium on particle-particle interactions and, thus, on the bulk solid’s cohesiveness, flowability and packing density.
Section 5.4. “Sintering” should be shortened. You refer several times to ref. [84], a text book on sintering phenomena, and highlight in this section also rather general and presumably to the audience well-known aspects of sintering.
Table 7, page 21: please clarify what you understand by average particle size. I guess you refer to the volume-averaged mean size, often referred as x50,3? I wonder, why you did not discuss or refer to works on the effect of width of the particle size distribution (PSD) on packing properties. I am not working in binder jetting, but in PBF processes, and there the aspect of width of PSD on powder spreadability, packing and processability is assessed in various papers and well-established.
Minor comments:
p. 2, l. 48: change “2D flies” to “2S files”
p. 5, l. 128: change “50 wt.% Zirconia” to “50 wt.% zirconia”
p. 5, l. 162: change “solutions and causing” to “solutions causing”
p. 8, l. 248: change “when thermally decompose and” to “when thermally decomposed and”
Author Response
Reviewer 2
Comments and Suggestions for Authors
Review on
“A Review on Binder Jet Additive Manufacturing of 316L Stainless Steel”
by
Mirzababaei and Pasebani
(manuscript jmmp- 508435)
Submitted June 2019 to “Journal of Manufacturing and Materials Processing”
In this paper the authors give an extensive review on binder jetting of 316L stainless steel. Up to date and relevant literature on materials and processing is considered in this well-written paper. The addressed topic is of great interest and in scope of the journal. I only have few remarks, which should be addressed by the authors in a revised version of the article. After incorporation of the proposed changes, the paper should be ready for publication. Below please find my remarks and suggestions for improvement.
Comments:
1. The discussion in section 4.1. “Powder Size and Shape” on bulk solids characteristics and spreadability, respectively, sinterability should be re-written and expanded. In the section, there are several flaws, for example, one cannot generalize that smaller particles have less surface roughness than larger one. Moreover, the discussion on the usage of bimodal distributed particles and wet vs. dry powder application seems somehow sloppy. In both cases, I would argue via the particle size dependency and dependency of the surrounding medium on particle-particle interactions and, thus, on the bulk solid’s cohesiveness, flowability and packing density.
Our Response:
The discussion is revised accordingly. Detailed explanations and discussion are added to section 5.1.1. Metal Powders and Powder Deposition in Binder Jetting.
Regarding bimodal/multimodal distributions and wet and dry deposition methods, the text is revised accordingly and more related explanations from useful articles are studied and added to the related section.
2. Section 5.4. “Sintering” should be shortened. You refer several times to ref. [84], a text book on sintering phenomena, and highlight in this section also rather general and presumably to the audience well-known aspects of sintering.
Our Response:
The sintering section is revised and shortened accordingly. Additional well-known BJ sintering data are added to related section (6.4.3. Sintering Kinetics at Intermediate and Final Stage).
Ref. [84] (currently [93]) as you mentioned is a useful sintering phenomena book. This work aims to more closely investigate the sintering mechanism (e.g., grain boundary diffusion) occurring during post-processing of BJ processes. Because reported work in the literature did not explicitly provide information on this area for BJ technology, this section is widely explained to provide enough information on detailed sintering processes. However, this revised one can be sufficient to express the purpose. The extra details and equations will provide in the next work when sintering is the focus of our work.
3. Table 7, page 21: please clarify what you understand by average particle size. I guess you refer to the volume-averaged mean size, often referred as x50,3? I wonder, why you did not discuss or refer to works on the effect of width of the particle size distribution (PSD) on packing properties. I am not working in binder jetting, but in PBF processes, and there the aspect of width of PSD on powder spreadability, packing and processability is assessed in various papers and well-established.
Our Response:
Yes, the average in this text means ‘Mean’. The related section is revised accordingly.
Regarding width of PSD, related data are reported in subsection 7.1. (P. 18 l. 612-616)
4. Minor comments:
p. 2, l. 48: change “2D flies” to “2S files”
p. 5, l. 128: change “50 wt.% Zirconia” to “50 wt.% zirconia”
p. 5, l. 162: change “solutions and causing” to “solutions causing”
p. 8, l. 248: change “when thermally decompose and” to “when thermally decomposed and”
Our Response:
All of these minor comments are revised accordingly.
Reviewer 3 Report
This paper is just a collection of well-known articles published in Binder jet manufacturing. I doubt that there is a paper here, it is written as a broad overview with no particular emphasis. A good review paper tries to tell a story suing previously published work to illustrate the message. However, I was unable to find any particular emphasis/message in this article. It seems simply a summary of the previously published literature. In its current form, it is not acceptable. What is the link between section 1.1 and 1.2? Why section 3.1 as there is no 3.2 section available in the manuscript? What is the new knowledge developed in this paper?
Author Response
Reviewer 3
1. This paper is just a collection of well-known articles published in Binder jet manufacturing. I doubt that there is a paper here, it is written as a broad overview with no particular emphasis. A good review paper tries to tell a story suing previously published work to illustrate the message. However, I was unable to find any particular emphasis/message in this article. It seems simply a summary of the previously published literature. In its current form, it is not acceptable. What is the link between section 1.1 and 1.2? Why section 3.1 as there is no 3.2 section available in the manuscript? What is the new knowledge developed in this paper?
Our Response:
The novelty and contribution of this work as well as gap in the literature is added as a new section (section 2.) and is added to summary to explicitly address these issues.
The link between section 1.1 and 1.2. is added by few relating sentences. Also, to make it a better structure both subsections are merged. Currently section 1. has no subsection.
Section 3. (Materials) which is currently section 4. is revised accordingly. Several lines of explanations are added to the text, and subsections are numbered accordingly. Currently there are three subsection for section 4. As mentioned in the first answer, an additional section (section 2.) is added to express the developed knowledge of this work. This is also reflected in the summary.
Currently, no review article has summarized and compared the binder jetting manufactured SS 316L. This paper for the first time, reviews, summarizes and compares all the BJ of SS 316L and highlights the existing gaps in the literature in section 2 and summary.
Round 2
Reviewer 3 Report
The authors made no efforts to address the reviewer comments. This paper still looks like a collection of summaries of cited articles. A review article generally has a thesis statement and the storyline flow around the statement. Also, it generally articulates a future research direction. I could not find anything like that. The authors added to section 2 regarding the current status of Binder jet technology without any references. In contrast, in the other sections, it appears that the authors tried to increase the number of references. An example, a single statement cannot have 12 references [1-12]. There are so many references without saying particularly what they did, why they did like that and what was the outcome of the previous study.
The reason why a review article differs from a summary article that a review article does not have redundancy. Just an example powder size affects the surface roughness, higher final density of the printed product, mechanical properties and dimensional accuracy. However, the effects of powder size appeared in sections 5.1.1, 5.2.1,7.1, 7.6 in this article. there are many more redundancies like this need to be addressed.
I cannot agree with the statement: "The authors of this review article highlight the existing gaps in the literature and provide a comprehensive summary and comparison in the field of BJ, in particular, SS 316L."- I cannot find anything like that. It is either not highlighted proper or diluted with so many summary statements. There are many more statements like this that needs to be addressed properly.
I cannot find any reason for third order many subsections such as 5.1.1, 5.1.2 etc. They can be reduced to second order.
This kind of statement is not suitable for a journal article "To the best knowledge of authors, there has not been any review paper published on BJ of metallic parts. Thus, it is the novelty of this article to provide such a comprehensive review of BJ of SS 316L that correlates powder feedstock properties, printing parameters, and post‐processing parameters to part density, microstructure, and mechanical properties." This can be used as a response to reviewers comments not in the article itself.
